

# Tropospheric HONO Distribution and Chemistry in the Southeast U.S.

Chunxiang Ye[1,2], Xianliang Zhou[2,3], Dennis Pu[3], Jochen Stutz[4], James Festa[4], Max Spolaor[4], Catalina Tsai[4], Christopher Cantrell[5], Roy L. Mauldin III[5, 6], Andrew Weinheimer[7], Rebecca S. Hornbrook[7], Eric C. Apel[7], Alex Guenther[8], Lisa Kaser[7], Bin Yuan[9, 10], Thomas Karl[11], Julie Haggerty[7], Samuel Hall[7], Kirk Ullmann[7], James Smith[7,12], John Ortega[7]

[1] College of Environmental Sciences and Engineering, Peking University, Beijing, 100871, China

[2] Wadsworth Center, New York State Department of Health, Albany, NY

[3] Department of Environmental Health Sciences, State University of New York, Albany, NY

[4] Department of Atmospheric and Oceanic Sciences, University of California, Los Angeles, CA

[5] Department of Atmospheric and Oceanic Sciences, University of Colorado-Boulder, Boulder Colorado

[6] Department of Physics, University of Helsinki, Helsinki, Finland

[7] National Center for Atmospheric Research, Boulder, Colorado

[8] Department of Earth System Science, University of California, Irvine, CA

[9] NOAA, Earth System Research Laboratory, Chemical Sciences Division, Boulder, Colorado

[10] Cooperative Institute for Research in Environmental Sciences, University of Colorado at Boulder, Boulder, CO, USA

[11] Institute of Atmospheric and Cryospheric Sciences, University of Innsbruck, Innsbruck, Austria

[12] University of Eastern Finland, Kuopio, Finland

Correspondence to: Chunxiang Ye (c.ye@pku.edu.cn) and Xianliang Zhou (xianliang.zhou@health.ny.gov)



**Abstract**
Here we report the measurement results of nitrous acid (HONO) and a suite of relevant
parameters on the NCAR C-130 research aircraft in the Southeast U.S. during NOMADSS
2013 summer field study. Daytime HONO concentrations ranged from low parts per trillion
by volume (pptv) in the free troposphere (FT) to mostly within 5 - 15 pptv in the background
terrestrial air masses, and to up to 40 pptv in the industrial and urban plumes in the planetary
boundary layer (PBL). There was no discernable vertical HONO distribution trend in the PBL
above the lowest flight altitude of 300 m, indicating that the ground surface HONO source
was not a significant contributor to the HONO budget in the measurement altitude between
300 m and 4.7 km. While there was a strong correlation between the concentrations of HONO
and oxides of nitrogen ($NO_x = NO + NO_2$) ($R^2 = 0.52$), the sum of all known $NO_x$-related
HONO formation mechanisms was found to account for less 20% of the daytime HONO
source in the background terrestrial air masses, due to the low level of $NO_x$ and surface area
density of aerosol particles. Photolysis of particulate nitrate ($pNO_3$) appeared to be the major
daytime HONO source in the background terrestrial air masses, based on the measured $pNO_3$
concentration and the median value of $2.0 \times 10^{-4}$ s$^{-1}$ for $pNO_3$ photolysis rate constant
determined in the laboratory using ambient aerosol samples collected during the field study.
Within the power plant and industrial plumes encountered, daytime HONO was
predominantly produced by secondary formation processes involving both $NO_x$ and $pNO_3$ as
precursors. While HONO was not a significant OH precursor compared to $O_3$ under low $NO_x$
conditions in the air column, it was an important intermediate product of a photochemical
renoxification process recycling nitric acid and nitrate back to $NO_x$. Finally, the HONO/$NO_x$
ratio stayed relatively constant for several hours after sunset in the nocturnal residual layer,
suggesting no significant night-time volume HONO source existed in the nocturnal residual
layer and the nocturnal FT under background conditions.
**1   Introduction**

Extensive field studies at ground sites have shown that gas-phase nitrous acid (HONO)

exists at much higher levels than expected during the day, with a mixing ratio of HONO up to
several parts per billion by volume (ppbv) in the urban atmosphere (Acker et al., 2006;
Villena et al., 2011) and up to several hundred parts per trillion by volume (pptv) in rural
environments (Acker et al., 2006; Kleffmann et al., 2003; Zhang et al., 2009; Zhou et al.,
2002, 2011). At the observed concentrations, HONO photolysis (R1) becomes an important or



even a major OH primary source in both urban (Elshorbany et al., 2010; Villena et al., 2011)
and rural environments near the ground surface (Acker et al., 2006; He et al., 2006;
Kleffmann et al., 2003; Zhou et al., 2002, 2011).
$$HONO + h\nu \rightleftharpoons OH + NO \qquad\qquad (R1, R\text{-}1).$$
The OH radical is responsible for the removal of primary pollutants, and plays a crucial role in
the formation of secondary pollutants, such as $O_3$ and aerosol (Finlayson-Pitts and Pitts,
2000), and thus HONO, as an important OH precursor, plays an important role in atmospheric
chemistry.

The removal processes of HONO from the troposphere are relatively well understood,

including mainly photolysis, reaction with the OH radical and surface deposition. Photolysis
is the dominant sink for HONO during the day (Kleffmann et al., 2003; Oswald et al., 2015;
Zhang et al., 2009, 2012), and dry deposition is the major HONO loss pathway at night,
especially over wet surfaces (He et al., 2006; VandenBoer et al., 2015). However, HONO
sources in the planetary boundary layer (PBL) are numerous. HONO is directly emitted from
combustion processes, such as automobile emissions (Li et al., 2008b) and biomass burning
(Burling et al., 2010; Trentmann et al., 2003). Due to the relatively short photolytic lifetime of
HONO, in the order of 10 min around summer noontime, the impacts of the direct emission
on HONO distribution and chemistry is highly localized and limited to the source region
during the day. Recent studies have suggested that microbial activities produce nitrite through
nitrification or denitrification in the soil, and soil emission may be a significant HONO source
for the overlying atmosphere ( Maljanen et al., 2013; Oswald et al., 2013; Su et al., 2011).
Since the emission of HONO from soils depends on multiple factors, such as the abundance of
soil nitrate and ammonia, the soil pH and water content, and microbial types and activities, it
is expected that the strength of this HONO emission varies greatly in different environments
and thus needs to be further quantified (Oswald et al., 2013).

HONO is a unique species that is produced through heterogeneous reactions of

different precursors, such as $NO_2$ and $HNO_3$, on surfaces (R2 - R3):
$$NO_2 + H_2O \text{ (or organics)} \xrightarrow{\text{surface}} HONO \qquad\qquad (R2)$$
$$HNO_3(s) + h\nu \xrightarrow{\text{organics,H}_2\text{O}} HONO + NO_2 \qquad\qquad (R3)$$
Heterogeneous reactions of $NO_2$ with organics (R2) on the surfaces have been found to be
greatly accelerated by sunlight through photosensitization (George et al., 2005; Kleffmann,
2007; Stemmler et al., 2006, 2007) and these reactions are likely the major daytime HONO




source in urban environments (Acker et al., 2006; Villena et al., 2011; Wong et al., 2011).
Laboratory studies have confirmed that $HNO_3$ undergoes photolysis in sunlight at rates 2 - 3
orders of magnitude greater on the surface than in the gas phase (Baergen and Donaldson,
2013; Du and Zhu, 2011; Ye et al., 2016a, b; Zhou et al., 2003; Zhu et al., 2008), producing
$NO_x$ and HONO. In low-$NO_x$ environments, photolysis of nitric acid/nitrate deposited on the
surface has been proposed to be the major daytime HONO source near the ground surface (Ye
et al., 2016b; Zhou et al., 2003, 2011).
Several processes within an air mass may lead to volume, or *in situ*, production of
HONO. The OH+NO reaction (R-1) in the gas phase may be a significant HONO source in
high $NO_x$ and photochemically reactive atmospheres (Kleffmann, 2007; Villena et al., 2011),
but becomes negligible in low-$NO_x$ environments(Li et al., 2014; Ye et al., 2016b). Two
additional gas-phase reactions have been also proposed to produce HONO within the air
column:  between excited $NO_2$ ($NO_2^*$) and water vapor (R4) (Li et al., 2008a), and between
$NO_2$ and the hydroperoxyl-water complex ($HO_2 \cdot H_2O$) (R5a) (Li et al., 2014):
$$NO_2^* + H_2O \rightarrow HONO + OH \tag{R4}$$
$$HO_2 \cdot H_2O + NO_2 \xrightarrow{\alpha} HONO + O_2 + H_2O \tag{R5a}$$
$$HO_2 \cdot H_2O + NO_2 \xrightarrow{1-\alpha} products \tag{R5b}$$
However, further laboratory evidence suggests that reaction (R4) is too slow to be important
(Carr et al., 2009; Wong et al., 2011). And recent airborne observations have demonstrated
that the HONO yield ($\alpha$) from reaction (R5) is less than 0.03 (Ye et al., 2015).
Almost all HONO measurements to date have been made at ground stations. The
observed HONO concentrations reported in the literature represent the HONO levels in the
lower PBL under the significant but varying influence of ground surface processes. Thus, it is
difficult to distinguish the ground surface HONO sources from the *in situ* HONO sources.
Measurements of the vertical profile of HONO concentrations and/or HONO fluxes have
suggested that ground surfaces can be major HONO sources for the overlying atmosphere in
many cases (He et al., 2006; Kleffmann et al., 2003 Stutz et al., 2002; Zhou et al., 2011), but
not in some other cases (Villena et al., 2011). A recent HONO flux measurement has
suggested that the HONO source from the forest canopy contributed ~ 60% of the measured
HONO budget at the measurement height of 11 m above the forest canopy, and the *in situ*
HONO production contributed the remaining ~ 40% (Zhou et al., 2011). Similarly,
observational and modeling studies implied a presence of a volume HONO source at 130-m





altitude above Houston, TX (Wong et al., 2012, 2013). The relative importance of *in situ*
HONO production would be expected to increase with altitude due to decreasing influence of
the ground surface, at least during the day. Airborne measurements in the air mass above the
altitude influenced directly by ground HONO sources should provide more direct and
quantitative evidence for *in situ* HONO production in the troposphere. Indeed, the limited
number of airborne measurements available have shown that HONO exists in substantial
amounts throughout the troposphere ( Li et al., 2014; Ye et al., 2015; Zhang et al., 2009).

Here we report airborne HONO measurement results and findings from five research

flights in the Southeast U.S. during the NOMADSS (Nitrogen, Oxidants, Mercury and Aerosol
Distributions, Sources and Sinks) 2013 summer field campaign aboard the NSF/NCAR C-130
research aircraft.
**2   Experimental**
NOMADSS was an airborne field study under the "umbrella" of SAS (Southeast Atmosphere
Study). It consisted of nineteen research flights on board the NSF/NCAR C-130 aircraft from
June 1, 2013 to July 15, 2013. Parameters observed included HONO, $HNO_3$, particulate
nitrate, $NO_x$, $O_3$, BrO, OH radicals, $HO_2$ radicals, $RO_2$ radicals, aerosol surface area densities
(size <1 μm), VOCs, photolysis frequencies, and other meteorology parameters. Table 1
summarizes the instrumentation, time resolution, detection limit, accuracy, and references for
the measurements.

HONO was measured by two long-path absorption photometric (LPAP) systems based

on the Griess-Saltzman reaction (Zhang et al., 2012; Ye et al., 2016b).  Briefly, ambient air
was first brought into the aircraft through an inlet and then HONO was scrubbed using de-
ionized (DI) water in a 10-turn glass coil sampler to ensure high efficiency HONO sampling.
The scrubbed nitrite was then derivatized with 5 mM sulfanilamide (SA) and 0.5 mM N-(1-
Naphthyl)-ethylene-diamine (NED) in 40 mM HCl, to form an azo dye within 5 min. The azo
dye was detected by light absorbance at 540 nm using an optic fiber spectrometer (LEDSpec,
WPI) with a 1-m liquid waveguide capillary flow cell (WPI). "Zero-HONO" air was
generated by directing the sample stream through a $Na_2CO_3$-coated denuder to remove HONO
and was sampled by the systems periodically to establish measurement baselines. Interference
from $NO_x$, PAN, and particulate nitrite if any, was corrected by subtracting the baseline from
the ambient air signal. Due to the low collecting efficiency of these interfering species in the
sampling coil and their low concentrations, the combined interference was estimated to be less




than 10% of the total signal. Potential interference from peroxynitric acid ($HO_2NO_2$) was
suppressed by heating the PFA sampling line to 50 °C with a residence time of 0.8 s. The
$HO_2NO_2$ steady state concentration was estimated to be less than 1 pptv at temperatures of 20
- 30 °C in the background PBL (Gierczak et al., 2005), and thus interference from $HO_2NO_2$
was negligible. Whereas in power plant plumes and urban plumes in the PBL or biomass
burning plumes in the upper free troposphere (FT), $HO_2NO_2$ interference was not negligible.
HONO measurements were corrected by a term of "$0.2 \times [HO_2NO_2]_{SS}$", assuming an upper
limit $HO_2NO_2$-to-HONO conversion efficiency of 0.2 in our system. $[HO_2NO_2]_{SS}$ refers to the
steady state concentration of $HO_2NO_2$, and the upper limit $HO_2NO_2$-to-HONO conversion
efficiency of 0.2 was estimated from the ratio of the observed [HONO] to the calculated
$[HO_2NO_2]_{SS}$ in cold, high altitude air masses under our measurement conditions. In the PBL,
the correction is below 10% of the total signal. The accuracy of HONO measurements was
confirmed by comparison with a limb-scanning Differential Optical Absorption Spectroscopy
(DOAS) (Platt and Stutz, 2008). The agreement between these two instruments was very good
in wide power plant plumes where HONO mixing ratios significantly exceeded the detection
limits of both instruments (Ye et al., 2016b).
Particulate nitrate ($pNO_3$) was quantitatively collected with a frit disc sampler after a
NaCl-coated denuder to remove $HNO_3$ (Huang et al., 2002). The collected nitrate was reduced
to nitrite by a Cd column, and determined using a LPAP systems (Zhang et al., 2012). Zero air
was generated to establish measurement baselines for $pNO_3$ by passing the ambient air
through a Teflon filter and a NaCl-coated denuder to remove aerosol particles and $HNO_3$.
Potential interferences from HONO, $NO_x$ and PAN were corrected by subtracting the
baselines from the ambient air signals.
The mixing ratios of a large number of non-methane organic compounds (NMOCs)
were measured by Trace Organic Gas Analyzer (TOGA) (Hornbrook et al., 2011a) and
Proton-transfer-reaction mass spectrometry (PTR-MS) ( Karl et al., 2003; de Gouw and
Warneke, 2007). The surface area density of fine particles was measured by a Scanning
Mobility Particle Sizer (SMPS). The photolysis frequencies were determined by a Charged-
coupled device Actinic Flux Spectroradiometer instrument (CAFS) (Shetter et al., 2002). The
mixing ratios of $HO_x$ and $RO_2$ radicals were measured by a method based on selected-ion
chemical-ionization mass spectrometry (SICIMS) (Hornbrook et al., 2011b; Mauldin et al.,
2010). The mixing ratios of ozone and $NO_x$ were measured by NCAR's chemiluminescence



instruments (Ridley et al., 2004). Meteorology parameters were provided by state parameter
measurements on board the C-130.

The results from five out of nineteen flights are presented here to discuss vertical

HONO distribution and HONO chemistry in the Southeast U.S. The flight tracks are shown in
Figure 1.
**3 Results and Discussion**
**3.1 General data description**
Figure 2 shows the time series of HONO, $NO_x$, $pNO_3$ concentrations and the measurement
altitude for five selected research flights in the Southeast U.S. during the NOMADSS 2013
summer field study. Research flight (RF) #4, RF #5 and RF #17 are race track flights in the
background terrestrial areas designed to establish HONO distribution and explore HONO
chemistry in background air masses. RF #11 is a race track flight designed to intercept plumes
from local power plants and urban areas and explore HONO chemistry therein. All four flights
were conducted in the daytime, roughly from 14:00 to 22:00 UTC (10:00 to 18:00 EDT). RF
#18 is a race track flight conducted from 20:30 on July 12th to 03:30 on July 13th UTC (16:30
on July 12th to 00:30 on July 13th, 2013 EDT), aiming to study the potential night-time HONO
accumulation both in the PBL and the FT.

Table 2 summarizes the data statistics for HONO, $NO_x$ and $pNO_3$ measurements in the

PBL and the FT, and Figure 3 shows composite vertical distributions of HONO, $NO_x$ and
$pNO_3$ concentrations from the five flights in the Southeast U.S. during the NOMADSS 2013
summer field study. HONO, $NO_x$ and $pNO_3$ concentrations show horizontal gradients in every
race track flight and vary in different race track flights, reflecting the inhomogeneity of air
masses in the region. However, no significant vertical gradient in HONO, $NO_x$ and $pNO_3$
concentrations is apparent, which will be further discussed below. Except in a few power
plant plumes and urban plumes mostly encountered in RF #11 (labelled as A-G), most of the
data is representative of background terrestrial air masses. The range of the mixing ratio of
HONO is 1.1 – 35.9 pptv. The mean (±1SD) and median values of HONO concentration are
5.4 (±3.4) pptv and 4.2 pptv in the FT, and 11.2 (±4.3) pptv and 10.6 pptv in the PBL.
HONO levels at ~ 4 pptv are typically found in the background FT, but high HONO
concentrations up to 18.2 pptv are also observed in the elevated biomass burning plumes.
Many biomass burning plumes were observed during other flights and will be discussed in a
future paper. HONO levels at ~ 11 pptv are representative of background conditions in the



PBL. High HONO levels up to 35.9 pptv are observed in the power plant plumes and urban
plumes in RF #11. The HONO distribution and chemistry in these urban and power plant
plumes in the Southeast U.S. are specifically discussed below, in comparison with the results
for background conditions (RF # 4, #5, and #17). These measured HONO values are
consistent with the range of 4 – 74 pptv in the troposphere over Northern Michigan (Zhang et
al., 2009), but are significantly lower than other airborne observations (up to 150 pptv) in the
morning residual layer over an industrial region of Northern Italy (Li et al., 2014), where the
levels of HONO precursors, such as $NO_x$ and $pNO_3$, were much higher.
The range of the mixing ratio of $NO_x$ is from several pptv to around 1.6 ppbv. The
mean ($\pm$1SD) and median values of $NO_x$ concentration are 96 ($\pm$52) pptv and 92 pptv in the
FT, and 313 ($\pm$174) pptv and 278 pptv in the PBL. The mixing ratios of $NO_x$ are mostly
between 50 - 150 pptv in the background conditions in the FT and between 200 - 500 pptv in
the background conditions in the PBL. Similar to HONO, high values of $NO_x$ also occur in the
urban and power plant plumes in the PBL (up to 1.6 ppbv) and in the biomass burning plumes
in the FT (up to 0.6 ppbv).
Fewer measurement data points are available for $pNO_3$, compared to those for $NO_x$
and HONO, due to air bubble formation in the flow cell of the $pNO_3$ system, especially at
high altitudes. The range of the mixing ratio of $pNO_3$ is from 2 pptv to 216 pptv, with the
mean ($\pm$1SD) and median values of 28 ($\pm$25) pptv and 21 pptv in the FT, and 78 ($\pm$47)
pptv and 70 pptv in the PBL. The $pNO_3$ levels were highly variable in both the FT and the
PBL. In the FT, the $pNO_3$ levels were often under 10 pptv, but high concentrations up to 115
pptv were also observed in elevated biomass burning plumes. In the PBL, high $pNO_3$ levels
were sometimes observed in relative clean conditions; whereas, low $pNO_3$ levels were
observed in high HONO and $NO_x$ power plant plumes. Both the N(V) level (= [$HNO_3$] +
[$pNO_3$]) and the partitioning between $HNO_3$ and $pNO_3$ seem to play roles in determining the
$pNO_3$ level.
**3.2 HONO contribution from ground-level sources**
There are several ground-level HONO sources that may contribute to the HONO
budget in the overlying atmosphere. They include anthropogenic sources, such as power plant
and automobile emissions (Li et al., 2008b), and natural processes, such as soil emission
(Maljanen et al., 2013; Oswald et al., 2013; Su et al., 2011), heterogeneous reactions of $NO_2$
(Acker et al., 2006; George et al., 2005; Ndour et al., 2008, 2009; Ramazan et al., 2006) and




surface $HNO_3$ photolysis (Ye et al., 2016b; Zhou et al., 2003,2011). Since HONO photolytic
lifetime is relatively short, e.g. 8 - 16 min in RF #4, RF #5, RF #11 and RF #17, a steep
negative vertical gradient of HONO concentration would be expected if a significant
contribution originated from the ground. The lack of a significant vertical gradient in the
measured HONO concentrations (Fig. 3) thus suggests that the ground contribution is either
limited to the shallow layer of the boundary layer near the ground, below the C-130 lowest
flight altitude of 300 m, or small relative to the *in situ* production of HONO in the air column
(Ye et al., 2017).

To further examine the potential HONO contribution from the ground sources, vertical

profiles of HONO, $NO_x$, and $pNO_3$, are compared with those of potential temperature (K) and
isoprene measured, for example, in the first race-track of RF#4 from 11:00 – 12:15 LT (Fig.
4). Indeed, the measurements conducted in the PBL from 300 m to 1200 m were above the
unstable surface layer, as indicated by the constant potential temperature (Fig. 4e). The
vertical distribution of isoprene originating from the ground can be expressed with the
following equation (Eq.1):

$$\ln\left(\frac{C}{C_0}\right) = -\frac{k\tau}{H}h = -\frac{h}{h^*}$$    (Eq. 1)

where, $C_o$ and $C$ are its concentrations near the ground and at the altitude $h$, $k$ is the pseudo-
first order degradation rate constant, $H$ is the boundary layer height, $\tau$ is the average mixing
time in the PBL, and $h^*$ ($= H/(k\tau)$) is its characteristic transport height within one degradation
lifetime of isoprene. According to the best fit of (Eq.1) to the observed isoprene data (Fig. 4d),
its characteristic transport height $h^*$ is estimated 692 m for isoprene. Assuming isoprene is
mainly oxidized by the OH radical whose average concentration is estimated at $3 \times 10^6$ mole
$cm^{-3}$ in the PBL (Kaser et al., 2015), the pseudo-first order degradation rate constant of ~
$3.0 \times 10^{-4}$ $s^{-1}$ (or the degradation rate of ~ 0.93 $h^{-1}$) is determined for isoprene. Based on a
boundary layer height of ~1.2 km (Fig. 4e), an average PBL mixing time $\tau$ is estimated to be
~1.6 h between 11:00 – 12:15 LT of RF #4. With a photolytic lifetime of ~ 11 min for HONO,
the estimated characteristic transport height of HONO is 138 m between 11:00 – 12:15 LT in
RF #4, well below 300 m, the lowest flight altitude of the C-130 aircraft during this field
study. Therefore, the instrument on-board the C-130 would not detect the HONO contribution
from the ground sources during this race-track profiling around noontime. However, it is
interesting to note that there was a slight increase in HONO concentration at the two lowest
altitudes (Fig. 4a), which may be attributed to the increasing concentrations of its precursors,
$NO_x$ and $pNO_3$ (Fig. 4b, c), both which are much longer lived than HONO.





Apart from the rapid photolytic loss of HONO, the rate of vertical mixing plays an
important role in limiting the transport height of HONO in the PBL. The vertical mixing of
the PBL is enhanced from the morning to the afternoon, as the ground surface is heated by
solar radiation gradually during the day. The average mixing time in the PBL is reduced from
~ 3 h in the morning, to ~ 1.5 h around noontime, and to ~ 30 min in the afternoon,
determined from isoprene gradients from RF #4, #5 and #17. The characteristic transport
height of HONO would be ~ 500 m in the afternoon, i.e., some of the ground emitted HONO
could survive and be transported to lower measurement altitudes, and thus may be detected by
our profile measurements. However, the contribution from ground HONO sources to the
observed HONO concentrations in the PBL above 300 m appear to be limited, as indicated by
the lack of consistent vertical HONO gradient above the altitude of 300 m (Fig. 3a) in all the
race track flights

**3.3 Daytime HONO chemistry in low $NO_x$ areas**

After removing the data measured in the urban and power plant plumes, the daytime HONO
concentrations are mostly within the range of 5 - 15 pptv throughout the PBL in the
background terrestrial areas in the five race-track research flights. Photolysis of HONO is its
dominant sink, with a photolysis lifetime of 8 - 16 min during these four daytime flights (RF
#4, RF #5, RF #11, and RF #17). Therefore, there must be a significant volume HONO
source, up to 200 pptv $h^{-1}$, within the air mass to sustain the observed HONO concentrations.
Both $NO_x$ and $pNO_3$ are potential HONO precursors in the air column. Figure 5 shows
the correlation analysis of HONO with $NO_x$ and $pNO_3$ in the background terrestrial air masses
during the five flights. While HONO correlates relatively well with $NO_x$ ($r^2 = 0.52$), with a
fitted HONO/$NO_x$ ratio around 0.04, it only weakly correlates with $pNO_3$ ($R^2 = 0.14$) (Fig. 5).
It may appear at first that $NO_x$ is a more important HONO precursor than $pNO_3$. However, the
detailed analysis below suggests that $NO_x$ is only a minor precursor to the observed HONO,
and photolysis of $pNO_3$ is the major *in situ* HONO source.
The photo-stationary state HONO concentration ([*HONO*]$_{pss}$) was calculated using
Equation 2 that takes into account all the known HONO source contributions from $NO_x$-
related reactions, including gaseous reactions of OH and NO (R-1), excited $NO_2$ ($NO_2^*$) and
water vapor (R4) (Carr et al., 2009; Li et al., 2008a), $NO_2$ and the hydroperoxyl-water
complex ($HO_2 \cdot H_2O$) with an upper limit HONO yield of 3% (R5a)(Li et al., 2014; Ye et al.,
2015), and heterogeneous reaction of $NO_2$ on aerosol surfaces (R2) using an upper limit





uptake coefficient of $10^{-4}$ reported in the literature (George et al., 2005; Monge et al., 2010;
Ndour et al., 2008, 2009; Stemmler et al., 2006, 2007):
$$[HONO]_{pss} = \frac{k_{-1}[NO][OH]+k_4[NO_2^*][H_2O]+\alpha k_5[NO_2^*][H_2O]+k_2 S_{aerosol}[NO_2]}{J_{HONO}+k_{OH-HONO}[OH]}$$  (Eq. 2)
where $S_{aerosol}$ is the aerosol surface area density. Under typical daytime conditions in the PBL
with the median measured values of reactants, the upper limit $[HONO]_{pss}$ value is less than 2
pptv, much lower than the median measured HONO concentration of ~ 11 pptv. Figure 6a
shows the relationship ($r^2 = 0.44$) between the photolytic HONO loss rate with the sum of
HONO production rates from all the $NO_x$-related reactions calculated with upper-limit
reaction rate constants. A slope of about 0.19 indicates that the contribution from these $NO_x$-
related reactions to the volume HONO source is minor in the background troposphere, despite
the good correlation between HONO and $NO_x$. The high HONO/$NO_x$ ratios up to 0.24 in the
low-$NO_x$ air masses are indicative of more important contributions from other HONO
precursors, such as $pNO_3$.
Photolysis of $HNO_3$ on surfaces has been found to proceed at a much higher rate than
in the gas phase (Baergen and Donaldson, 2013; Du and Zhu, 2011; Ramazan et al., 2004; Ye
et al., 2016b; Zhou et al., 2003; Zhu et al., 2008), with HONO as the major product on
environmental surfaces (Ye et al., 2016a, 2017). Furthermore, photolysis of particulate nitrate
has been found to be the major daytime HONO source in the marine boundary layer (Ye et al.,
2016b). To examine the role of particulate nitrate as a potential HONO source in the
troposphere, aerosol samples over the terrestrial areas were collected on Teflon filters on
board the C-130 aircraft during the NOMADSS 2013 summer field study and were used in the
light-exposure experiments to determine the photolysis rate constants for particulate nitrate in
the laboratory. The determined $pNO_3$ photolysis rate constant ($J_{pNO_3}^N$) varies over a wide
range, from $8.3 \times 10^{-5}$ s$^{-1}$ to $3.1 \times 10^{-4}$ s$^{-1}$, with a median of $2.0 \times 10^{-4}$ s$^{-1}$ and a mean ($\pm 1$
standard deviation) of $1.9 (\pm 1.2) \times 10^{-4}$ s$^{-1}$, when normalized to tropical noontime conditions
at ground level (solar zenith angle = 0 °), and the average HONO to $NO_2$ relative yield is 2.0
(Ye et al., 2017). Figure 6b shows the relationship between the photolytic HONO loss rate
($J_{HONO} \times [HONO]$) and the volume HONO production rates from $pNO_3$ photolysis ($2/3 \times J_{pNO_3}$
$\times [pNO_3]$). The median $J_{pNO_3}^N$ of ~ $2.0 \times 10^{-4}$ s$^{-1}$ was used to calculate the ambient $J_{pNO_3}$ by
scaling to $J_{HNO3}$:
$$J_{pNO_3} = J_{pNO3}^N \times \frac{J_{HNO_3}}{7.0 \times 10^{-7} \, s^{-1}}$$  (Eq. 3),





where $J_{HNO_3}$ is the photolysis rate constant of gas-phase $HNO_3$ calculated from light intensity
measurement on the C-130 aircraft, and $7.0 \times 10^{-7}$ s$^{-1}$ is the photolysis rate constant of gas-
phase $HNO_3$ under the tropical noontime condition at ground level (solar zenith angle = 0 °). A
slope of 0.67 can be derived from Figure 6b, suggesting that $pNO_3$ photolysis is the major
volume HONO source. However, the r$^2$ of 0.31 is not as strong as expected from $pNO_3$
photolysis being the major volume HONO source. The lower than expected correlation
coefficient may be due to the fact that only a single median $J_{pNO3}^{N}$ value of $\sim 2.0 \times 10^{-4}$ s$^{-1}$ is
used in the calculations of the ambient $J_{pNO_3}$ and the production rates of HONO in Figure 6b,
while the actual $pNO_3$ photolysis rate constants determined from seven NOMADSS aerosol
samples are highly variable, ranging from $8.3 \times 10^{-5}$ s$^{-1}$ to $3.1 \times 10^{-4}$ s$^{-1}$ (Ye et al., 2017). The
production rates of HONO in Figure 6b are thus only rough estimates of the *in situ* HONO
production rates from $pNO_3$ photolysis in different air masses.

HONO photolysis has been found to be an important or even a major OH primary

source in the atmosphere near the ground surface ( Elshorbany et al., 2010; He et al., 2006;
Kleffmann et al., 2003; Villena et al., 2011; Zhou et al., 2011). However, HONO is not a
significant daytime OH precursor in the background troposphere away from the ground
surface. Based on the measurement results in this study, the contribution of HONO photolysis
to the OH source budget (mean ± SD) is 52 ± 22 pptv h$^{-1}$ in the PBL and 28 ± 20 pptv h$^{-1}$ in
the FT, respectively, less than 10% of the OH production contributed by $O_3$ photolysis.
However, since HONO is mainly produced from photolysis of particulate nitrate, it becomes
an important intermediate product of a photochemical renoxification process recycling nitric
acid and nitrate back to $NO_x$. The regenerating rate of $NO_x$ of about 52 pptv h$^{-1}$ via HONO
photolysis is equivalent to an air column $NO_x$ source of $\sim 2 \times 10^{-6}$ mol m$^{-2}$ h$^{-1}$ in the 1- km
PBL, a considerable supplementary $NO_x$ source in the low-$NO_x$ background area.

### 3.4 HONO chemistry in plumes

One of the objectives of RF #11 was to study the chemistry of HONO in urban and coal fired
power plant plumes. The arrows and corresponding labels in Figures 2 and 7 indicate the
urban plumes (A – C) and power plant plumes (D - G). Benzene was used as the tracer of
urban plumes (Liu et al., 2012; Shaw et al., 2015).  Benzene peaks were observed in all urban
plumes (A – C), but not in the power plant plumes (D – G). The power plant plumes were
generated from high-intensity point sources, and thus had features of narrow but high peaks of
both HONO and $NO_x$ concentrations in the time-series plots (Figs. 2 and 7). In contrast, the



urban plumes were generated from area sources and thus were shown as broad peaks of
HONO and $NO_x$ in the time-series plot with low levels of $NO_x$ (mostly below 500 pptv) (Fig.
2). There were a few sharp but small $NO_x$ peaks within the broad urban plumes, reflecting the
contributions of some point sources in the urban areas. The observed $HONO/NO_x$ ratio was
around 0.02 in the power plant plumes, lower than that of ~ 0.05 in urban plumes and in
background terrestrial air masses. Based on the distances between measurement locations
from the power plants or the centre of urban area and the observed wind speed, the transport
times of these power plant plumes were estimated to be ≥1 h, over 5 times longer than HONO
photolysis lifetime of 8 - 16 min. Therefore, most of the observed HONO in the power plant
plumes was produced *in situ* within the air masses. Since the typical emission ratio of
$HONO/NO_x$ is less than 0.01 in the fresh power plant plumes and automobile engines
(Kurtenbach et al., 2001; Li et al., 2008b), the elevated $HONO/NO_x$ ratios observed in the
plumes suggest the presence of other HONO precursors, such as $pNO_3$.

Figure 7b shows the time-series plot of HONO budget within the air masses sampled

by the C-130 aircraft during flight RF# 11, comparing its photolysis loss rate with its
production rates from $pNO_3$ photolysis and from all the $NO_x$-related reactions combined.
Photolysis of particulate nitrate appears to be the major volume HONO source in all urban
plumes and in most of the power plant plumes except for plume G observed here (Fig. 7b).
$NO_x$ was generally more important as a HONO precursor in the power plant plumes than in
the urban plumes and in low-$NO_x$ background terrestrial air masses, due to higher levels of
$NO_x$ (up to 1.6 ppb in Fig. 7a), OH radical and aerosol surface density. For example, all the
$NO_x$-related reactions combined contributed up to 52% of the total volume HONO source
required to sustain the observed HONO concentration in plume G (Fig. 7b). In fresh power
plant plumes encountered during the RF #7 to Ohio River Valley (X. Zhou, unpublished data),
over 20 ppb $NO_x$ was detected, and the $NO_x$-related reactions were found to account for
almost all the required HONO source to sustain the observed HONO. The power plant plumes
undergo rapid physical and photochemical evolution during the day, such as dilution and
$NO_x$-into-$HNO_3$ conversion. Thus, the relative contributions from $NO_x$-related reactions and
particulate nitrate photolysis as HONO sources change rapidly as the plumes age.

### 382    3.5 Night-time HONO chemistry

HONO accumulation near the ground surface during the nighttime has been widely observed
(Kleffmann et al., 2003; Oswald et al., 2015; 2008; Stutz et al., 2002, 2010; VandenBoer et



al., 2013, 2014, 2015), contributed by various anthropogenic and natural HONO sources on
the ground. The main objective of RF #18 was to study the night-time HONO evolution in
both the nocturnal residual layer and the nocturnal FT. After sunset, the surface cooling
promotes the formations of a inversion layer near the ground surface and a nocturnal residual
layer above; the contribution from ground HONO sources then becomes negligible to the air
masses beyond the surface inversion layer.  Meanwhile, no effective HONO sinks, such as
photolysis, oxidation by OH and dry deposition, exist in the nocturnal residual layer. Thus the
HONO accumulation, if any, is a net contribution from dark heterogeneous $NO_2$ reaction on
aerosol surfaces (R2).

The C-130 flew in an elongated race track pattern along a north-south direction, about

140 km from Nashville, TN (Fig. 1), alternating between the PBL (1200 m) and the FT (2500
m), from late afternoon to midnight local time (Fig. 2). In the FT, HONO and $NO_x$
concentrations were relatively stable throughout the afternoon and the night, staying around 4
ppt and 90 pptv respectively. The lack of night-time HONO accumulation is expected from
the low levels of HONO precursors, mostly $NO_2$, and surface area of aerosol particles in the
FT (Fig. 2).

The conditions in the PBL were far more variable and complicated. There were strong

horizontal gradients of $NO_x$, $pNO_3$ and HONO in the PBL, with higher concentrations at the
southern end and lower concentrations at the northern end of the flight track. Back-trajectory
analysis using NOAA's HYSPLIT model (Stein et al., 2015) indicates that the encountered air
masses in the PBL at the southern end passed over Nashville, about 140 km northeast of the
sample area, with a transport time of about 6 h (Fig. 8a), while the air masses at the northern
end stayed to north of Nashville (Fig. 8b). Therefore, the anthropogenic emissions from the
metropolitan area of Nashville contributed to the higher concentrations of pollutants observed
at the southern end of the flight track. There were also trends of increasing concentrations of
$NO_x$, $pNO_3$ and HONO with time after the sunset (Fig. 2).  This was probably a result of less
dispersion and dilution of anthropogenic pollutants, including $NO_x$, as the PBL became more
stable after sunset. Furthermore, as time progressed from late afternoon into evening and
night, the air masses were less photochemically aged during the transport from the source
areas, due to the decreasing solar light intensity and shorter solar light exposure time.

Because of the large spatial and temporal variations in the concentrations of HONO and

its precursors in the PLB (Fig. 2), it is difficult to directly evaluate the nighttime HONO
accumulation from HONO measurements alone. The concentration ratio of HONO and its



dominant nighttime precursor, $NO_2$, can be used as an indicator of nighttime HONO
accumulation. As the air masses at measurement altitude of 1200 m decoupled from the
ground-level processes after sunset, the HONO production from heterogeneous $NO_2$ reaction
(R2) on aerosol surface becomes the only HONO source, and can be expressed by the
following equations (Eq. 4 and Eq. 5):
$$P(HONO) = \frac{1}{4} \times \left[\frac{s}{v}\right] \times \sqrt{\frac{8RT}{\pi M}} \times \gamma \times [NO_2] \qquad \text{(Eq.4)}$$
$$\frac{P(HONO)}{[NO_2]} = \frac{1}{4} \times \left[\frac{s}{v}\right] \times \sqrt{\frac{8RT}{\pi M}} \times \gamma \qquad \text{(Eq.5)}$$
where $\left[\frac{s}{v}\right]$ is the specific aerosol surface area density, $R$ is the gas constant, $K$ the absolute
temperature, $M$ the molecular weight of $NO_2$, and $\gamma$ is the dark uptake coefficient of $NO_2$
leading to HONO production. The $NO_2$-normalized HONO accumulation over time, $\Delta\frac{[HONO]}{[NO_2]}$,
can then be calculated by equation (Eq. 6):
$$\Delta\frac{[HONO]}{[NO_2]} \sim \frac{1}{4} \times \left[\frac{s}{v}\right] \times \sqrt{\frac{8RT}{\pi M}} \times \gamma \times \Delta t \qquad \text{(Eq. 6)}$$
Assuming a dark uptake coefficient $\gamma$ of $1 \times 10^{-5}$ of $NO_2$ on aerosol (George et al., 2005;
Monge et al., 2010; Ndour et al., 2008; Stemmler et al., 2006, 2007) with a $\left[\frac{s}{v}\right]$ value of $\sim 10^{-4}$
$m^{-1}$, a relative HONO accumulation rate, $\Delta\frac{[HONO]}{[NO_2]}/\Delta t$ of $\sim 0.0003$ $h^{-1}$ is estimated using the
equation (Eq. 6), equivalent to a HONO accumulation of 0.13 pptv $hr^{-1}$ at a constant $NO_2$
concentration of 400 pptv. Such a low HONO accumulation rate is below our measurement
detection limit. Indeed, the calculated HONO to the $NO_x$ ratio using the measurement data
stayed almost unchanged with time (Fig. 9), well within the observational variability after the
sunset, suggesting no significant volume production of HONO in the nocturnal boundary
layer.

## 4    Conclusions

Substantial levels of HONO existed during the day in both the PBL (median ~ 11 pptv) and
the FT (median ~ 4 pptv) over the Southeast U.S. during the NOMADSS 2013 summer field
study. It appears that ground HONO sources did not significantly contribute to the HONO
budget in the PBL above the minimum measurement heights of 300 m. Photolysis of
particulate nitrate was the major volume HONO source in the background low-$NO_x$ air
masses, while $NO_x$ was only a minor HONO precursor. Up to several tens pptv of HONO



were observed in coal fired power plant plumes and urban plumes during the day; the major
HONO precursor could be either $NO_x$ or $pNO_3$ depending on the chemical characteristics and
photochemical age of the plumes. No significant night-time HONO accumulation was
observed in the nocturnal residual layer and the free troposphere, suggesting no significant
night-time volume HONO source due to low levels of $NO_x$ and specific aerosol surface area.
HONO was not a significant daytime OH precursor in the rural troposphere away from the
ground surface; however, HONO mainly produced from photolysis of particulate nitrate could
significant provide a renoxification pathway. The $NO_x$ regeneration rate of about 52 pptv h$^{-1}$
in rural PBL is a considerable supplementary $NO_x$ source in a low-$NO_x$ background region.
**Acknowledgements**
This research is funded by National Science Foundation (NSF) grants (AGS-1216166, AGS-
1215712, and AGS-1216743). We would like to acknowledge operational, technical, and
scientific support provided by NCAR, sponsored by the National Science Foundation. The
data are available in our project data archive
(http://data.eol.ucar.edu/master_list/?project=SAS). Any opinions, findings, and conclusions
or recommendations expressed in this paper are those of the authors and do not necessarily
reflect the views of NSF.





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





Table 1. Measurements from the NOMADSS 2013 summer study used in this analysis.

| Parameters | Instrument | Time Resolution | Detection Limit | Accuracy | References |
|---|---|---|---|---|---|
| HONO | LPAP | 200 s | 1 pptv | 20% | (1, 2) |
| pNO$_3$ | LPAP | 360 s | 2 pptv | 30% | (1, 2, 3) |
| HNO$_3$ | LPAP | 20 min | 2 pptv | 30% | (1, 2, 3) |
| NO | CI | 1 s | 20 pptv | 10% | (4) |
| NO$_2$ | CI | 1 s | 40 pptv | 15% | (4) |
| O$_3$ | CI | 1 s | 100 pptv | 5% | (4) |
| OH | SICIMS | 30 s | *$5\times10^4$ | 30% | (5, 6) |
| HONO | DOAS | 60 s | ~ 30 pptv | 20% | (7) |
| Photolysis Frequencies | CAFS | 6 s | | 10-15% | (8) |
| Surface area density | SMPS/UHSAS | 65 s/1 s | | 20% | (9) |
| VOCs | PTRMS | 15 s | | 20% | (10, 11) |
| VOCs/organic nitrates | TOGA | 20 s | | 20% | (12) |

*in molecules cm$^{-3}$
LPAP: long-path absorption photometric (LPAP) systems
CI: 4-channel chemiluminescence instrument
SICIMS: selected-ion chemical-ionization mass spectrometer
DOAS: Differential Optical Absorption Spectroscopy
CAFS: Charged-coupled device Actinic Flux Spectroradiometer
SMPS: Scanning Mobility Particle Sizer
UHSAS: Ultra-High Sensitivity Aerosol Spectrometer
PTRMS: Proton Transfer Reaction Mass Spectrometry
TOGA: Trace Organic Gas Analyzer
References: (1) Zhang et al., 2012; (2) Ye et al., 2016b; (3) Huang et al., 2002; (4) Ridley et
al., 2004; (5) Hornbrook et al., 2011b; (6) Mauldin et al., 2010; (7) Platt and Stutz, 2008;
(8) Shetter et al., 2002; (9) Flagan, 2011;   (10) Karl et al., 2003; (11) de Gouw and
Warneke, 2007; (12) Hornbrook et al., 2011a.





Table 2. Data statistics for HONO, $NO_x$ and $pNO_3$ measurements both in the PBL and the FT
from the five Southeast U.S. research flights during the NOMADSS 2013 summer field study.

|  |  | HONO, pptv | $NO_x$, pptv | $pNO_3$, pptv |
|---|---|---|---|---|
| **PBL** | Range | 3.1 - 35.9 | 81 - 1635 | 7 - 216 |
|  | Mean ± SD | 11.2 ± 4.3 | 313 ± 174 | 79 ± 47 |
|  | Median | 10.6 | 278 | 70 |
| **FT** | Range | 1.1 - 18.2 | <10 - 582 | 2 - 115 |
|  | Mean(±SD) | 5.4 ± 3.4 | 96 ± 52 | 28 ± 25 |
|  | Median | 4.2 | 92 | 21 |







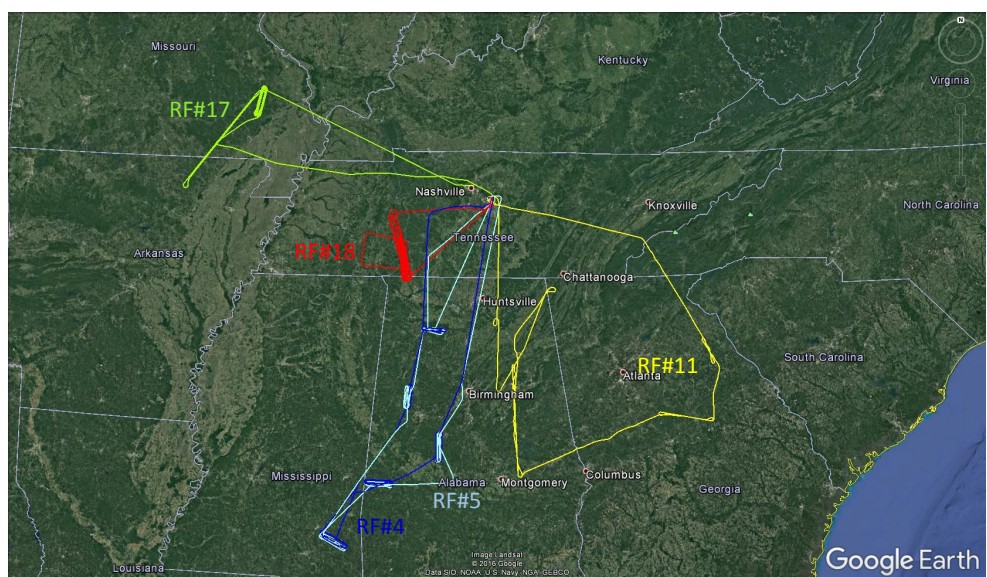

Figure 1. Flight tracks in the Southeast US during the NOMADSS 2013 summer study. The
flight start time and end time in UTC (= EDT+4)) are: RF#4 (blue): 15:12 and 22:30, June 12,
2013; RF#5 (light blue): 15:04 and 21:52, June 14, 2013; RF#11 (yellow): 15:20 and 21:02,
June 29, 2013; RF#17 (green): 15:07 and 21:57, July 11, 2013; RF#18 (red): 20:32, July 12,
2013, and 03:37, July 13, 2013



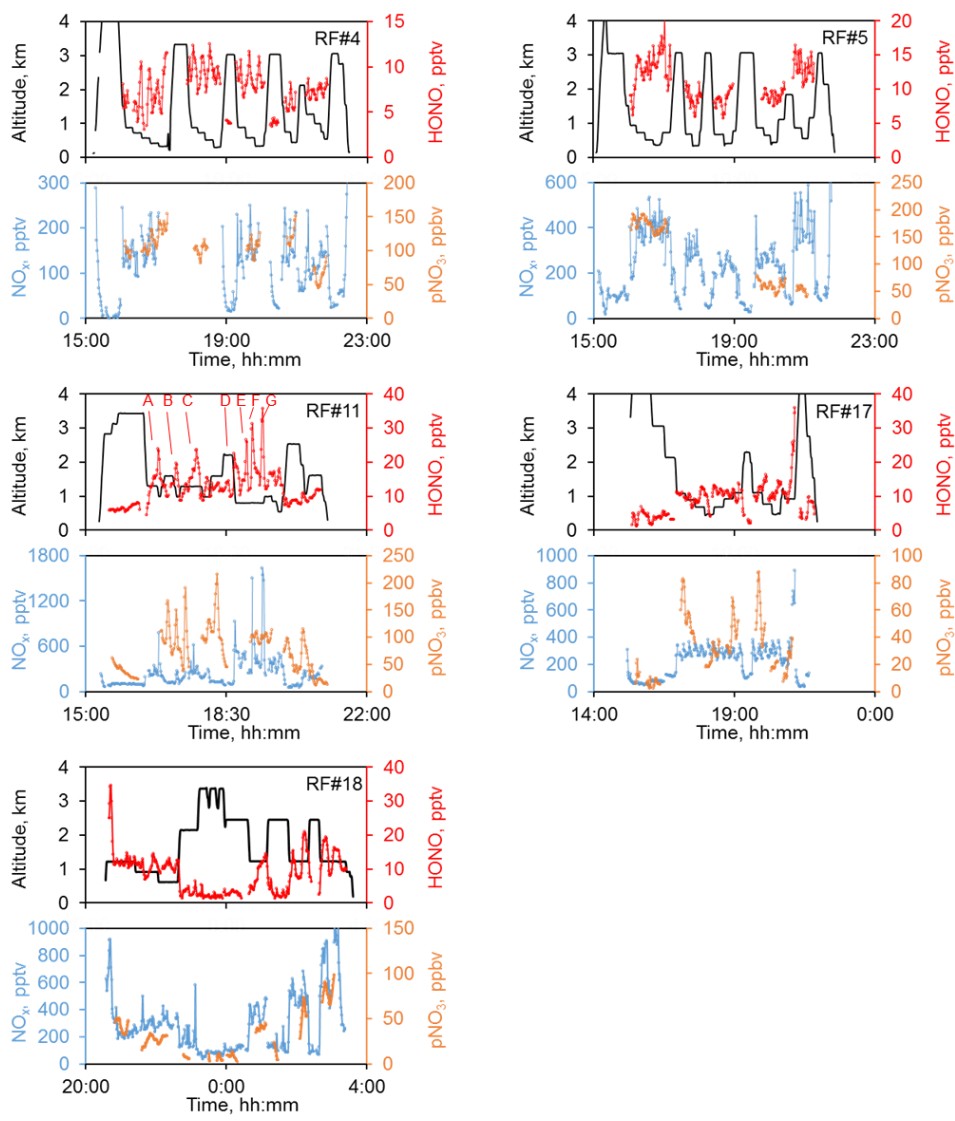


Figure 2. Time series of altitude, HONO, $NO_x$ and $pNO_3$ in five flights (RF #4, RF #5, RF
#11, RF #17 and #18) in the Southeast US during the NOMADSS 2013 summer study. In RF
#11, A-C indicate urban plumes, and D-G indicate coal-fired power plant plumes. The time is
in UTC.




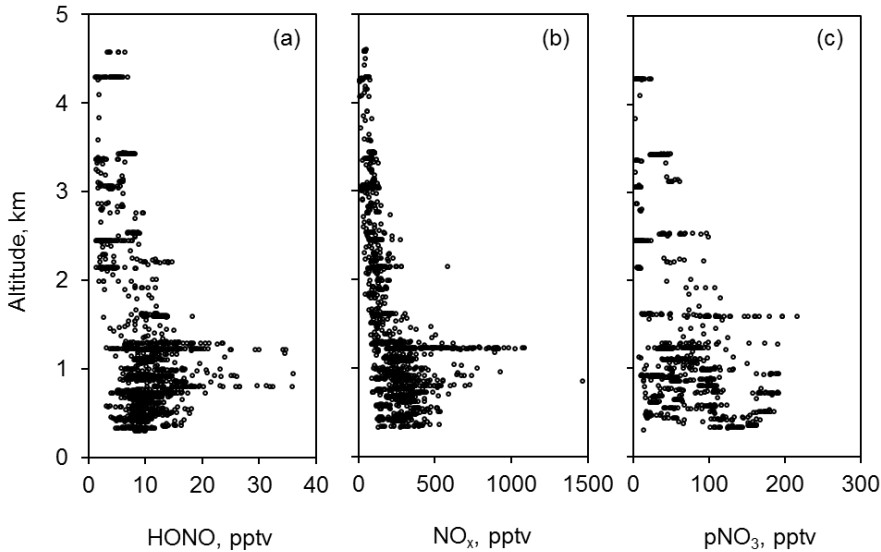


Figure 3. Vertical distributions of concentrations of HONO (a), NO$_x$ (b), and pNO$_3$ (c) in the
five selected flights in the Southeast US during the NOMADSS 2013 summer study.







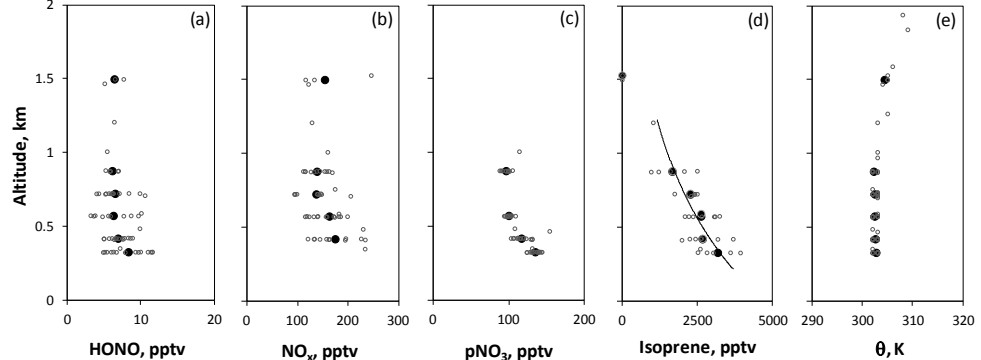


Figure 4. Vertical distributions of concentrations of HONO (a), $NO_x$ (b), $pNO_3$ (c), isoprene
(d) and potential temperature (e) in the PBL during the first race-track of RF#4 from 11:00 –
12:15 LT (16:00 – 17:15 UTC), June 12, 2013. The small open circles represent the 1-min
data points, the large solid circles the mean values for each race-track measurement altitude.
The line in (d) is the best fit of (Eq. 1) to the isoprene data: $h = 5.97 - 0.692 \ln C$, $r^2 = 0.93$.






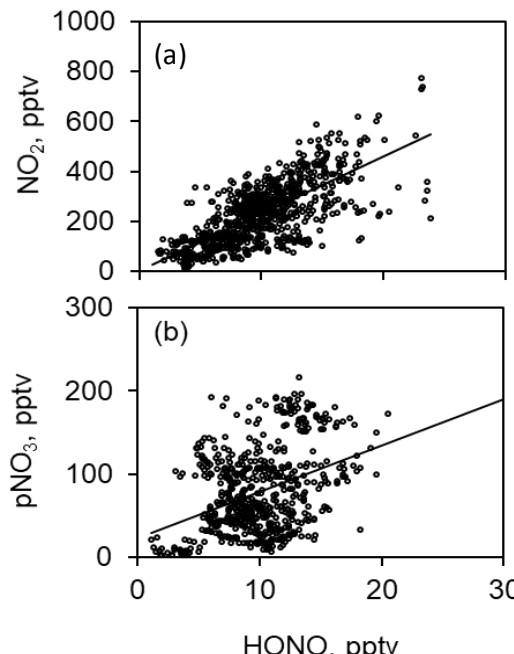



Figure 5. Correlation analysis of HONO with $NO_x$ (a, $r^2$=0.52) and $pNO_3$ (b, $r^2 = 0.14$) in the
southeast US during the NOMADSS 2013 summer study. Data points in the urban and power
plant plumes have been excluded.








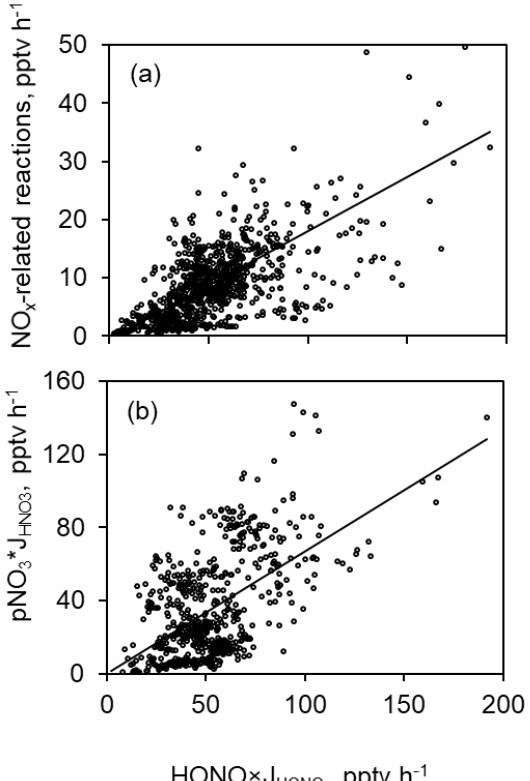


Figure 6. Correlation analysis of main HONO sink ("HONO×$J_{HONO}$") with contribution from
particulate nitrate photolysis, pNO3×$J_{pNO3}$ (a) and with contribution from NO$_x$ related
reactions (b) in the southeast US during the NOMADSS 2013 summer study. The line
represents the least-squares fitting ($R^2$=0.44, intercept = -0.57 and slope = 0.19 for Figure 6a;
$R^2$=0.31, intercept = 0.05 and slope = 0.67 for Figure 6b).






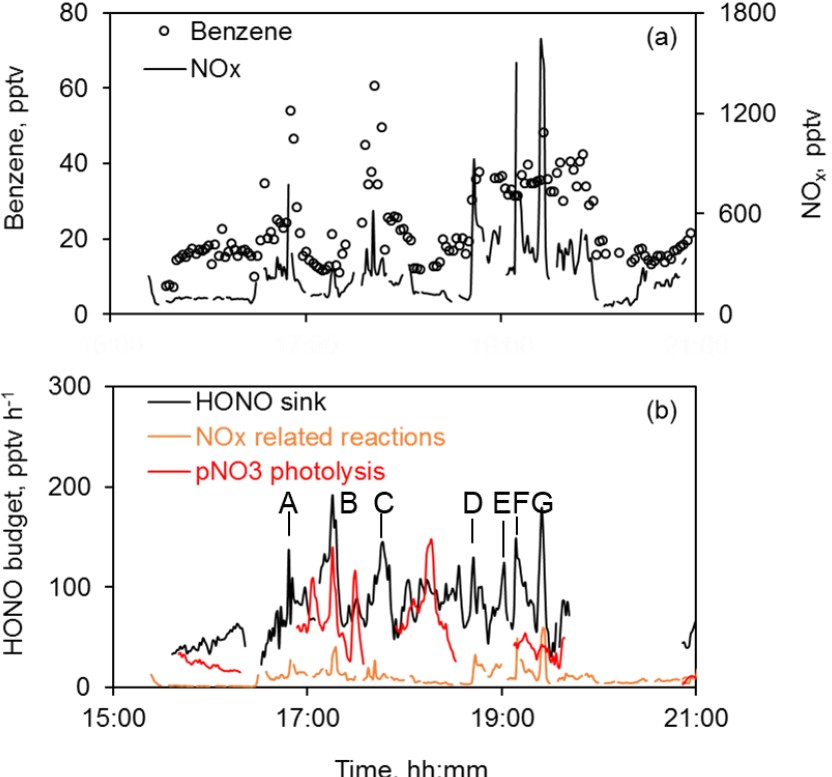


Figure 7. HONO budget analysis in RF #11 in the Southeast US during the NOMADSS 2013

summer study. "HONO sink" is the HONO loss rate contributed by photolysis and the

reaction of HONO with OH radicals, "NOx related reactions" is the sum of HONO

productions by all known $NO_x$ reactions, and "pNO3 photolysis" is the HONO source

contributed by photolysis of $pNO_3$





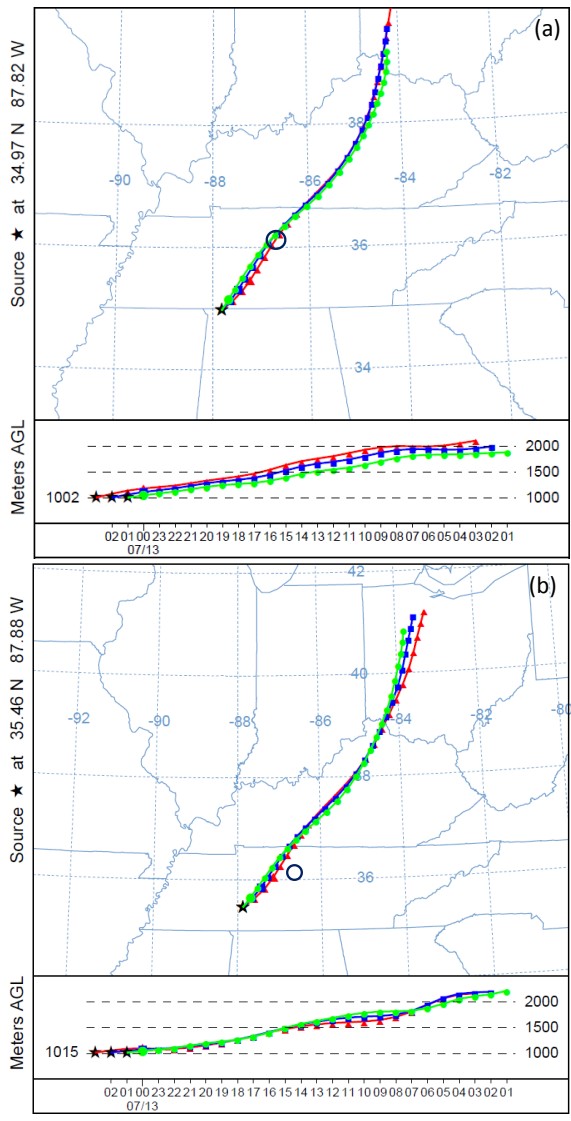


Figure 8. Back trajectory analysis of air masses encountered in the PBL in RF #18 in the
Southeast US during the NOMADSS 2013 summer study. The air masses arriving at the
southern point of the flight tracks were found to pass over the metropolitan area of Nashville
(the black circle, panel a), while those at the northern point to stay to the north of the area.
The back trajectory analysis was made using NOAA's online HYSPLIT model
(http://www.arl.noaa.gov/HYSPLIT_info.php).





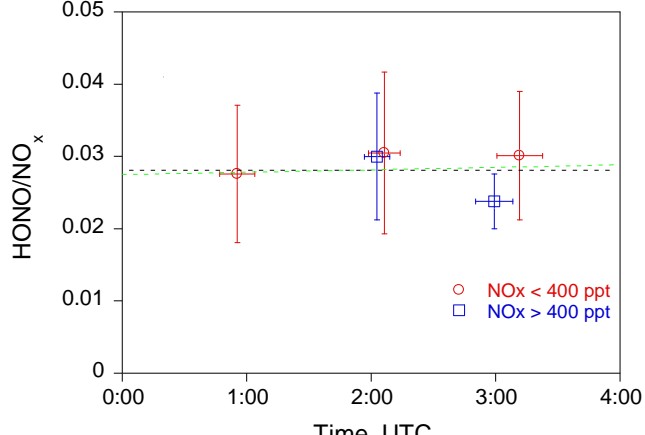


Figure 9. The evolution of HONO/NO$_x$ ratio in the nocturnal boundary layer during the
RF#18. The red circles and blue squares are the median HONO/NO$_x$ values under the
conditions of NO$_x \leq$ 400 pptv and NO$_x >$ 400 pptv, respectively. The horizontal bars indicate
the averaging time periods and the vertical bars the one standard deviation of HONO/NO$_x$
ratios. The black dashed line is the least squared fit to the data, and the green dashed line
indicates a slope of $3\times10^{-4}$ hr$^{-1}$. The sunset time at the sampling location was 0:40 UTC.