# Peer review of "Tropospheric HONO Distribution and Chemistry in the Southeast U.S."

_Atmospheric Chemistry and Physics, 2018_

## Short Comment (SC1) · 2 Mar 2018

The manuscript "Tropospheric HONO Distribution and Chemistry in the Southeast U.S" by Ye et al. presents airborne measurements of reactive nitrogen compounds in the troposphere. They measure HONO to be larger than can be explained by known formation processes and find that known mechanisms explain only 20% of the daytime HONO source in background air masses. Understanding HONO formation and loss is important to understanding the photochemistry of the atmosphere, but the results here require further support to be useful in constraining reactions that produce HONO. Some specific concerns are detailed below.

1) The discussion of the measurements and their uncertainties are insufficient, and

many of the experimental descriptions are qualitative. Substantially greater quantitation is required to support the stated 1 ppt detection limit. For example, zeros were performed "periodically" (line 125), and the baselines were subtracted from the total signal. How frequently were these backgrounds performed, and how was the background determined outside of the zero periods? Was a single value used for a flight, or was the background determined by interpolating between zeroes? The inlet residence time of 0.8 s is very large. What happens in a NOx plume? Wouldn't there be a contribution from NO2 conversion to HONO on the inlet? A description of the inlet length and flow would be helpful. If the HONO measurement is a difference between total signal and background, I am surprised that there are no values below zero in Figures 2 and 3. Are there really never any instances when HONO falls to zero? Perhaps the interferences are underestimated.

Please mention briefly how surface area density was determined from SMPS data. Wouldn't SMPS also provide a constraint on aerosol mass that could be useful for verifying the pNO3 measurements? Some of values of pNO3 in remote regions are very large - up to 0.5 ug/m3 and sometimes comparable to NOx, so it would be useful to have other measurements to support this. Have similarly large nitrate values been measured outside of urban plumes over the SE US in other studies?

I cannot find the mentioned UHSAS data or DOAS data in the project archive. What does a "very good" agreement mean (line 142)? Again, quantifying the agreement and showing data would strengthen the paper.

Why is OH estimated using a prior study (line 246), when the OH measurements listed in Table 1 could be used?

2) A very large photolysis rate for pNO3 is used to explain HONO formation, but this rate isn't consistent with the data shown. It is difficult for me to understand the difference between "determined photolysis rate" and "ambient photolysis rate" (section 3.3), but both are extremely large and comparable to the loss rate for isoprene. The nitrate

photolysis rates give a nitrate lifetime of approx. one hour, which is less than the lifetime of NOx. How can nitrate ever accumulate in the atmosphere if its lifetime is so short? Are there any other studies that find a very short lifetime for nitrate? The large nitrate photolysis rate is inconsistent with the nitrate abundance and distribution reported here and cannot explain the HONO abundance.

3) The different air mass types are not explained, and it isn't clear if or how they were separated. Benzene is used to identify urban plumes, but how are power plant plumes and biomass burning plumes identified? Could there be a large biomass burning plume contribution to the observations? Were some plumes a combination of sources? CO or acetonitrile measurements could be used to identify air mass influences. Similarly, SO2 was measured and could be used to identify power plant plumes. I could find no mention of any meteorological conditions. Without a more thorough description of the ambient conditions and ancillary measurements, it is very difficult to compare these results with other studies.

The large reduction in PBL mixing time (line 262) between noon and afternoon is very surprising and differs from previous studies. By noon in the summer, the mixing time should be much less than 1.5 h.

4) Relevant literature is not referenced, and the differences with previous measurements are not discussed. We published a very similar paper, using aircraft HONO measurements at the same time and location and under the same SAS umbrella (Neuman at al., HONO emission and production determined from airborne measurements over the Southeast U.S., JGR, 2016), but oddly, that paper is not referenced. We found that known HONO production mechanisms explained the HONO abundance, and we did not need to invoke unknown sources. In contrast, the studies referenced in the introduction (lines 29-30, line 103) report much larger values ranging from 100s of pptv to ppb levels. Why do the HONO values reported here differ from previous measurements, which range from indistinguishable from zero to ppbv levels? Meaningful comparisons to previous studies (some conducted at the same time and location) are

essential for understanding the findings reported here.

Line 85 states that nearly all HONO measurements have been made at ground sites, but that dismisses the many studies of HONO vertical gradients using DOAS and from towers (e.g. Young et al, Vertically Resolved Measurements of Nighttime Radical Reservoirs in Los Angeles and Their Contribution to the Urban Radical Budget, ES&T, 2012; Stutz et al., Simultaneous DOAS and mist-chamber IC measurements of HONO in Houston, TX, Atmospheric Environment, 2010; Vandenboer 2013 in the references). And the authors themselves have many papers that detail airborne measurements.

5) smaller points I don't know what an N(V) level is (line 216)

Data averaging is not explained. The time resolution of HONO and pNO3 are listed as 3 min and 6 min, yet 1 min data are shown. How are the data averaged in figure 3? The values do not match those shown in Figure 2, but the binning and averaging are never described.

Figure 2 shows pNO3 in ppbv, which is in error.

––––––––––––––––––––––––––––––

---

## Short Comment (SC2) · 13 Mar 2018

"Extraordinary claims require extraordinary evidence" is a phrase made popular by Carl Sagan [Rational Wiki]. This is particularly relevant to the Ye et al. [2018] paper. The authors make the extraordinary claim that "...the sum of all known NOx-related HONO formation mechanisms was found to account for less 20% of the daytime HONO source in the background terrestrial air masses, ....". If this claim were true, it would possibly force a reassessment of our understanding of HOx and NOx budgets of the troposphere, depending on the details of the other 80% of the (unknown) sources. However, the evidence presented Ye et al., [2018] does not justify that claim. In fact, Neuman et al. [2016] conclude: "Outside of recently emitted plumes from known combustion sources, HONO mixing ratios measured several hundred meters above ground

level were indistinguishable from zero within the 15 parts per trillion by volume measurement uncertainty. The results reported here do not support the existence of a ubiquitous unknown HONO source that produces significant HONO concentrations in the lower troposphere." The conclusion of these two studies disagree strongly, yet the reported measurements were made from different aircraft with different instrumentation, but in the same region of the country over the same time period, summer 2013. Unfortunately, Ye et al., [2018] do not discuss or even cite Neuman et al. [2016]. The differences in the results reported in these two papers point to clear experimental problems in the measurements. Until these problems are resolved, the extraordinary claim of Ye et al. [2018] should not be published.

References

Neuman, J. A., et al. (2016), HONO emission and production determined from airborne measurements over the Southeast U.S., J. Geophys. Res. Atmos., 121, 9237–9250, doi:10.1002/2016JD025197.

Rational Wiki, https://rationalwiki.org/wiki/Extraordinary_claims_require_extraordinary_evidence

Ye, C., et al. (2018), Tropospheric HONO distribution and chemistry in the Southeast U.S., Atmos. Chem. Phys. Discuss., https://doi.org/10.5194/acp-2018-105.

---

## Referee Comment (RC1) · Anonymous Referee #1 · 31 Mar 2018

The manuscript 'Tropospheric HONO distribution and chemistry in the Southeast U.S.' by Ye et al. presents HONO measurements made during the NOMADSS campaign. The two main claims presented here are: (1) there is more HONO observed than can be explained by known chemistry, and (2) photolysis of particle nitrate accounts for this so-called missing HONO source. The analysis used to make both claims are weak, therefore, unconvincing. Moreover, the analysis on nighttime chemistry and production in power plant exhaust were hastily done and written. This work can be considered for publication only after significant improvements.

As for (1), the authors claim that because HONO photo-lifetime is 8 minutes, that direct emission of HONO can be disregarded. This is a misinterpretation of the concept of lifetime. Lifetime represents an e-folding time, meaning that ~36% of the original

amount still remains after 8 minutes since time zero, or time since emission. If HONO at the emission source is 10 ppb, approximately 50 to 60 minutes is required for HONO to reach 11 ppt, the median HONO level reported. Judging by figure 2, the median HONO value of 11 ppt (what the authors claim is anthropogenic-free HONO) is observed in close proximity to urban plumes (20 to 30 ppt, identified in figure 2). This 50 to 60 minute period is an underestimate since it does not account for re-formation of HONO by OH + NO, both of which are likely to be elevated in urban plumes. Bottom line is that the authors need to demonstrate convincingly that the 11 ppt HONO is not derived from anthropogenic sources (by quantitatively accounting for mixing, emissions, and chemistry), because the case for this so-called 'extra' HONO is the difference between 11 ppt and 2 ppt (amount of HONO expected assuming PSS without this 'extra' source). The analysis as it currently stands in inadequate.

As for (2), the authors conclude a causal relationship between photolysis of particulate nitrate and HONO based on rather weak correlation (figure 5). That is less than convincing. Moreover, a photolysis rate of 2e-4 sec-1 means a photo-lifetime less than 1.5 hours for particulate nitrates. What are these nitrates? inorganic or organic? Has there been any reports of particle-phase nitrates exhibiting photo-lifetimes on the order of 1.5 hours? Is there a mechanism proposed? What remains in the particle-phase as the nitrate is released as HONO? Is all of the nitrate turn into HONO, or NO or NO2 or HNO3? I am concerned the photolysis conducted in the laboratory is not atmospherically relevant. More information on this lab photolysis experiment may help. How do the nitrate abundances measured with this filter method compare to what other instruments (AMS? PILS?) have measured for particle nitrates?

And as for the power plant analysis, the same concerns I have for claim (1) applies here. You cannot assume just because the plume has been transported over 1 hour that none of the HONO observed is anthropogenic in origin. You need to know what the mixing ratio was near the emission point to know whether the HONO measured downwind was or was not directly emitted because the photo-lifetime is an efolding time, it does not just disappear after 8 minutes. Lastly, citing previous work on the subject would be useful. Recent work by Neuman et al. 2016 comes to mind (https://agupubs.onlinelibrary.wiley.com/doi/abs/10.1002/2016JD025197).
* * *

---

## Referee Comment (RC2) · Anonymous Referee #2 · 9 Apr 2018

General Comments

This manuscript explores the generation and fate of HONO above and within the planetary boundary layer over the southeastern United States during NOMADSS 2013 from several research flights aboard the NCAR C-130 aircraft. The vertical distribution of HONO throughout this layer is clearly demonstrated to be derived from volume sources, with a robust testing of the known mechanisms of HONO formation against parameterizations of particulate nitrate photolysis, which is emerging as an important source of tropospheric HONO. The Author's find that previously established volume-based mechanisms of HONO formation cannot account for the observed quantities and that the photolysis nitrate in the condensed phase can possibly explain the majority of the observed quantities. The Authors demonstrate that HONO is a minor OH source at

these altitudes when its production is driven solely from volume production and also that it is an important intermediate in the renoxification pathways of tropospheric transport of nitrogen oxides. Overall, this manuscript is well written with a solid analysis of the dataset. There are minor modifications necessary to make the manuscript more clear and concise in its purpose and findings. The removal of some figures and text by the production of a supporting information document would easily facilitate this.

Specific Comments

Page 2, Lines 7-10: The detailed analysis of the isoprene transport and subsequent lifetime calculations for HONO are a quantitative assessment of the decoupling of surface emissions from the observed HONO. The Authors should consider using their quantitative assessment as the basis for their statement here instead of the more qualitative observation of no vertical gradient.

Page 2, Lines 14-15: Please provide the average +/- SD of the actual fraction of the observed HONO that was generated by pNO3 photolysis from the presented calculations instead of 'appeared to be the major daytime HONO source'.

Page 2, Lines 20-25: Provide the quantitative findings from each section of the detailed analysis here over the general statements of relative importance. This will generate greater impact for this work.

Page 3, Line 39: Remove ', as an important OH precursor,' as it is redundant.

Page 3, Lines 51-57: I would suggest removing this length section and replacing it with a single sentence following the statements on combustion HONO sources (Line 48). This level of detail in the introduction is not really relevant to the tropospheric chemistry discussed in this work.

Page 4, Lines 75-84: The last two sentences demonstrate that R4 is unnecessary and it should likely be removed from here and from the presented data analysis, since it has been show to be a two-photon process. It should be removed here and the section on

the hydroperoxyl-water complex mechanisms should be replaced with one sentence on its existence and low yield of HONO.

Page 5, Lines 104-107: It would be useful to guide the readers through the major explorations of this dataset here. Consider listing the major sections of this work here in the order that they are presented in the abstract, manuscript, and conclusions, to improve clarity.

Page 5, Line 108: The experimental section could use subsections to improve clarity.

Page 5, Line 126: The baseline subtraction of interferences from particulate nitrite here does not acknowledge that there is a size-dependent collection efficiency in these style of instruments. For example, fog droplets would be effectively captured in the primary channel to appear as HONO and not be corrected for in the secondary channel. This has been demonstrated in other works with this analytical approach (e.g. (Sörgel et al., 2011) and references therein). Is there any potential for droplet nitrite interferences in these measurements where clouds may have been encountered?

Page 6, Lines 138-139: It is confusing to follow the logic of this estimation. Was the maximum possible interference determined in some sections of the dataset to set the limit at 0.2? If possible, add the quantitative approach used to a section in a Supporting Information document. If not, please improve the clarity here.

Page 6, Lines 142-144: Provide the correlation coefficient, slope, and intercept here to improve clarity and validity of analytical approach.

Page 6, Line 149: The order of the used apparatus is not clear. Presumably the denuder followed the filter? Please clarify.

Page 6, Line 160: Delete 'NCAR's'

Page 7, Line 161: What are 'state parameter measurements'?

Page 7, Lines 183-184: Remove this from here. It is discussed in sufficient detail later

and distracts from the results.

Page 7, Lines 186-188: Remove these statements. The information is already presented in the Table and does not need repeating.

Page 7, Lines 191-192: Delete the sentence on the future paper.

Page 8, Lines 194-196: Delete these and direct the reader to the relevant section at the end of the preceding sentence by adding '(Section 3.4)'

Page 8, Lines 201-203: Remove these statements. The information is already presented in the Table and does not need repeating.

Page 8, Lines 210-212: Remove these statements. The information is already presented in the Table and does not need repeating.

Page 9, Line 236: Here is the first definition of the altitudes considered to by the PBL versus the FT. The Authors should add their criteria for distinguishing between the PBL and FT to the methods section. If it would be a lengthy addition, then a condensed description with supporting details could be placed in the Supporting Information document.

Page 9, Lines 238-250: This is a fantastic analysis of the vertical mixing and transport of surface-emitted species, but it is outside the focus of this work. Consider relocating this detailed analysis to the Supporting Information document.

Page 9, Lines 250-254: Distinguish between ground-emitted and volume-produced HONO here to improve clarity.

Page 9, Line 256: 'of its precursors' should be 'of its potential precursors' since this work is yet to demonstrate this quantitatively (although it is shown quite well later).

Page 10, Lines 286-287: This was stated in the introduction as insignificant (and potentially invalid), so why have the authors chosen to include this in their analysis? Suggest removing throughout.

Pages 10-11, Lines 289-291: Consider providing a justification for selecting all upper limits in these calculations to improve clarity.

Page 11, Line 302: Remove ', such as pNO3.' As it is redundant for the transition between paragraphs.

Page 11, Lines 309-310: Remove 'over the terrestrial areas', 'on Teflon filters. . . summer field study'. This information is already presented in the methods.

Page 12, Lines 326-330: This is a single sentence and is difficult to follow. Consider breaking into 2-3 sentences to improve clarity.

Page 12, Line 331: Delete 'only rough'. Redundant. Also see comments on Figure 6 regarding weighted error analysis.

Page 13, Line 357: What is the error on this ratio of 0.02? Is it statistically different from the fresh power plant emissions?

Page 13, Line 370: Since plume G is the only case study from these labels, consider a uniform label for the urban emissions (A) and the remainder of the power plant plumes (B). The increasing lettered format makes it seem that each instance will be discussed.

Page 15, Line 439: The conclusions section of this manuscript is similarly qualitative, as the abstract is, despite the excellent quantitative analysis presented throughout the results and discussion. Suggest revisiting this section with more quantitative information to improve clarity and impact.

Page 25, Table 2: The +/- SD is in brackets in one part of the table and not the other. Please correct this. The terms PBL and FT are not defined anywhere in the manuscript and should be given at least an operational definition somewhere in the methods section. Lastly, the number of data points being used in each of these calculations should be provided in a column or in the caption.

Page 27, Figure 2: Consider moving this figure to the supporting information or removing it entirely from the manuscript. The only specific features necessary here are the plumes which are presented again in Figure 7. With respect to the urban and power plant plumes, it could be simpler to assign the urban plumes a single letter (such as A), and similarly assign all the power plant plumes with a single letter excepting the one plume discussed in detail, which could be assigned a third letter. With each plume having a different letter, the figure suggests that there is something different between these, when there is nothing in the discussion that suggests this is the case. It would improve the clarity to simplify this.

Page 28, Figure 3: This figure does not seem necessary for inclusion in the main manuscript and should be considered to be moved to the supporting information. Figure 4 and Table 2 provide redundant and better insight into the measurements.

Page 29, Figure 4: It could be useful to add the typical PBL to FT height as a shaded area (if it has some variability) or horizontal line in each panel to facilitate clarity between the figure data and the discussion.

Page 30, Figure 5: The two sentences in the paper communicate all the information contained in this figure. Suggest removing this figure altogether or relocating to the supporting information. Further, the correlation analysis undertaken here is unclear and may be subject to some error if an error-weighted analysis is not being used (Wu and Zhen Yu, 2018). Is the error in both datasets being taken into account when calculating the regression coefficient? Please update the analysis and discussion to reflect the approach and ensure it is robust for the presented data.

Page 31, Figure 6: The same regression questions from Figure 5 also apply here. Please clarify the approach utilized and ensure that the appropriate regression analysis and statistics have been used when interpreting the data.

Page 32, Figure 7: Panel (a) here can be move to the supporting information or removed altogether.
Page 33, Figure 8: This information in this figure is presented concisely in the discussion and the figure does not add anything further. Consider removing this figure from the manuscript.

Page 34, Figure 9: The lines are very hard to see on this figure and the green line does not print well. Suggest using two black lines that are thicker than those currently used, with different dashing to distinguish them. The markers are also defined by very thin lines that could be made thicker for clarity.

References

Sörgel, M., Trebs, I., Serafimovich, A., Moravek, A., Held, A. and Zetzsch, C.: Simultaneous HONO measurements in and above a forest canopy: Influence of turbulent exchange on mixing ratio differences, Atmos. Chem. Phys., 11(2), 841–855, doi:10.5194/acp-11-841-2011, 2011.

Wu, C. and Zhen Yu, J.: Evaluation of linear regression techniques for atmospheric applications: The importance of appropriate weighting, Atmos. Meas. Tech., 11(2), 1233–1250, doi:10.5194/amt-11-1233-2018, 2018.

---

## Author Comment (AC1) · 3 May 2018

C. Ye (c.ye@pku.edu.cn) and X. Zhou (xianliang.zhou@health.ny.gov)

*General comments: The manuscript "Tropospheric HONO Distribution and Chemistry in the Southeast U.S" by Ye et al. presents airborne measurements of reactive nitrogen compounds in the troposphere. They measure HONO to be larger than can be explained by known formation processes and find that known mechanisms explain only 20% of the daytime HONO source in background air masses. Understanding HONO formation and loss is important to understanding the photochemistry of the atmosphere, but the results here require further support to be useful in constraining reactions that produce HONO. Some specific concerns are detailed below.*

**Response:** We would like to thank Andy Neuman for his time and efforts in preparing this detailed and comprehensive comment. We have revised the manuscript accordingly to address his questions and concerns.   Specific concerns and questions are addressed below in this Response .

*Major concerns:*
*Q1: The discussion of the measurements and their uncertainties are insufficient, and many of the experimental descriptions are qualitative. Substantially greater quantitation is required to support the stated 1 ppt detection limit. For example, zeros were performed "periodically" (line 125), and the baselines were subtracted from the total signal. How frequently were these backgrounds performed, and how was the back- ground determined outside of the zero periods? Was a single value used for a flight, or was the background determined by interpolating between zeroes?*

**Response:** HONO measurement technique has been described in detail in the previous method paper (Zhang et al., 2012), therefore only brief description of the instrument was given in the manuscript to provide the key pieces of information (lines 117 – 144 in the original manuscript).   We have added significant amount of information to the revised manuscript as suggested.   To answer the reviewer's questions, more details are provided below; please refer to the method paper (Zhang et al., 2012) for instrumental details, such as HONO sampling, baseline substation and interference correction, nitrite derivatization, and absorbance measurement of azo dye derivative by LPAP technique.

  HONO was measured by two separate LPAP (long-path absorbance photometer) systems. Each system ran a 30-min measurement and zero cycle, with 20 min sampling ambient air and 10 min sampling "zero-HONO" air for baseline correction, and with a 15-min time offset between the two sampling cycles (Figure 1a). The "zero-HONO" air was generated by directing the ambient air stream through a $Na_2CO_3$-coated denuder to remove HONO while allowing most of interfering species ($NO_x$, PAN, and particulate nitrite) to pass through. The combination of overlapping ambient signals from the two systems provide a continuous HONO concentration measurement (Figure 1b, solid black circles).   The absorbance signals were sampled at a rate of 1 Hz (Figure 1a, blue and red circles), and were averaged into 1-min

or 3-min data (Figure 1b, blue and red circles, 1-min averaging).   The averaged signals were converted into concentrations based on calibration slope and air sampling and liquid flow rate information. The baseline correction was made by subtracting the ambient signals by the extrapolated line between the two adjacent stable "zero-HONO" air signals (Figure Sb, blue and red lines). The "zero-HONO" air signals were stable most of the time, and the slow drift in the baseline can be easily corrected for. The baseline was sometimes found to change rapidly in two circumstances: when the aircraft was transacting through a high $NO_x$ plume, and when the ambient pressure changed rapidly and significantly during the rapid ascending to or descending from high altitudes. For the first case, the interference from other reactive nitrogen species in high $NO_x$ plumes can be corrected by subtracting from the ambient air signals the increases in the "zero-HONO" air signals measured by the other HONO system. However, this correction was rarely needed, since the increases in the "zero-HONO" air signals were usually quite small even in the urban and power plant plumes (Figure 2). In the second circumstance, large baseline drifts were observed when the flow state of the scrubbing solution was disturbed by rapid pressure fluctuations. The up-shifting or down shifting of the baseline may result in over-correction or under-correction; the over-correction could then result in negative concentration numbers, as Andy Neuman pointed out. Fortunately, altitude changes in the PBL during the race-track profiling did not disrupt the liquid flow pattern enough to cause rapid baseline shift (Figure 1a). If the baseline shift was found to be caused by rapid pressure fluctuations and if reasonable baseline correction could not be made, the data points were excluded from analysis, regardless of the sign or magnitude of the data.

[Figure]

Figure 1. Time series of 1-Hz raw absorbance signals (blue and red circles) and flight altitude (black circles) (a) and of 1-min averaged absorbance signals ((blue and red circles) and calculated HONO concentrations (black solid circles) (b), during NOMADSS RF#4 on June 12, 2013.    The blue and red lines in panel (b) are the baselines extrapolated from the two adjacent "zero-HONO" air measurements.

[Figure]

Figure 2. One-hour time series of 1-Hz absorbance signals from two HONO systems (blue and red lines), 1-min averaged HONO (black circles) and $NO_x$ (green triangles with line) during NOMADSS RF11 on June 29, 2013. The blue and red arrows indicate the slight increases in the "zero-HONO" air signals due to potential interferences from $NO_x$, particulate nitrite and PAN in the power plant plumes.

The time resolution is defined as the 90% response time based on the signal transition from "zero-HONO" air to ambient air. The lowering of the flow rates of scrubbing and reagent solutions and increase in the length of liquid plumbing tubing resulted in a longer response time (200 s) compared to that reported for the ground-based system (110 s) (Zhang et al., 2012). The lower detection limit of the method was estimated to be ≤1 pptv, based on 3 times the standard deviation of the zero air signal (N >10).   An overall uncertainty of $\pm(1 + 0.2$ [HONO]) pptv was estimated, combining the uncertainties in signal acquisition and processing, air and liquid flow rates, standard preparation, and baseline correction. Again, the estimated overall uncertainty of $\pm(1 + 0.2$ [HONO]) pptv is significantly higher for the aircraft HONO measurements than that of $\pm(1 + 0.05$ [HONO]) pptv for the ground HONO measurements (Zhang et al., 2012), in part due to pressure fluctuation and baseline drifting on the aircraft.

*Q2: The inlet residence time of 0.8 s is very large. What happens in a NOx plume? Wouldn't there be a contribution from NO2 conversion to HONO on the inlet? A description of the inlet length and flow would be helpful.*

**Response:** Andy Neuman has made some fair comments regarding the long inlet deployed on the C-130 for HONO measurement. It was only during the instrument integration when we learned that there were exhaust vents next the inlet ports near our instrument location.   The vented aircraft cabin air might significantly contaminate our HONO measurement.   To avoid the potential contamination artifacts, the inlet port on the other side of the aircraft was used.   A heated 7-m long 3/8"-ID PFA inlet line was thus needed and used, and a high flow rate (210 L min$^{-1}$) ambient air was drawn by an auxiliary blower to reduce the air sample residence time in the inlet line. The resulting residence time in the inlet line is 0.14 s, not 0.8 s stated in the original manuscript.   We regret the error and have made the correction in the revised manuscript.

Our group has examined the potential interference from heterogeneous $NO_2$ reactions

on the inlet wall surface on HONO measurements many times and in different environments, and have found it not to be significant. Figure 3 shows the result of such an experiment conducted recently in downtown Albany.   HONO in the ambient air was measured by two HONO systems, one with a regular inlet, and the other with or without adding long piece of heated PFA tubing (10 m long, 1/4-OD and 1/8"-DI). At a sampling flow rate of 2 L min$^{-1}$, the residence time of air sample in the PFA tubing was ~2.4 s, about 17 times longer than 0.14 s for the aircraft systems. The ambient $NO_2$ concentration varied from ~1 to ~6 ppbv during the measurement period.   The comparison of the two time series by measured the two HONO systems shows no discernible difference within the estimated uncertainty, regardless if the extra long tubing was added or removed (Figure 3).

[Figure]

Figure 3. Ambient HONO concentrations measured by two HONO systems in Downtown Albany during April 19-20, 2016.   A 10-m PFA tubing (1/8"-DI) was added to the inlet of system 2 (red circles) from at 17:00 on April 19, and was removed at 7:53 on April 20, as indicated by the black arrow. The three black bars at 19:31 and 22:02 on April 19 and at 4:25 on April 20 indicate estimated measurement uncertainties at the measured concentrations.   The insert is the scatter plot of the measured HONO concentrations by the two systems.   The red symbols represent the measurements by the two systems with the same short inlets (10-cm long, 1/16"-ID), and the blue symbols represent the measurements when an additional 10-m tubing (1/8"-ID) was added to system 2. The line is the linear best fit for the data.

The accuracy of HONO measurements was also confirmed by comparison with a limb-scanning differential optical absorption spectroscopy (DOAS) (Ye et al., 2016a). When measuring in wide power plant plumes where HONO mixing ratios exceeded the lower detection limits of both instruments, the agreement between these two instruments was very good, within the assessed uncertainties (Extended Data Fig. 3 in Ye et al., 2016a).

*Q3: If the HONO measurement is a difference between total signal and background, I am surprised that there are no values below zero in Figures 2 and 3. Are there really never any instances when HONO falls to zero? Perhaps the interferences are underestimated.*

**Response:** The signals for the "zero-HONO" air were quite stable, and ambient signals were well above the baselines, even at the data sampling rate of 1 Hz (Figure 1a). As explained in the response to Q1, ambient signals are always higher than the baseline signals extrapolated from

the adjacent "zero-HONO" signals, except during the rapid ascending to and descending from high altitudes; large baseline drifts were observed when the flow state of the scrubbing solution was disturbed by rapid pressure fluctuations. Overcorrection of the upward-shifting baseline may sometimes result in negative values in HONO concentration. However, the data were excluded from analysis if the baseline shifts caused by rapid pressure fluctuations could not be reasonably corrected, regardless of the sign or magnitude of the data.

*Q4: Please mention briefly how surface area density was determined from SMPS data. Wouldn't SMPS also provide a constraint on aerosol mass that could be useful for verifying the pNO3 measurements? Some of values of pNO3 in remote regions are very large - up to 0.5 and Have similarly large nitrate values been measured outside of urban plumes over the SE US in other studies?*

**Response:** The surface area density was calculated by the following equation:

$$S/V = \sum (4\pi r_i^2) \times n_i$$

where $r_i$ and $n_i$ represent the radius and number density of aerosol particles. A perfect sphere was assumed for aerosol particle in the calculation.

The mean ($\pm 1$std) and median of $pNO_3$ in the Southwest US were 76 ($\pm 45$) pptv and 66 pptv in the PBL, and 35 ($\pm 39$) pptv and 15 pptv in the free troposphere, within the range of reported particulate nitrate in rural atmosphere (Heald et al., 2012 and paper therein). The high $pNO_3$ concentrations were observed in the PBL during the first racetracks of the RFs #4 and #5 west of Centreville, AL, and during the RF 11 around Auburn, AL (Figures 1 and 2). Agricultural activities in this region may release enough $NH_3$ to convert some of the gaseous $HNO_3$ into $pNO_3$, as observed by Neuman et al. (2003). We have calculated the aerosol mass using the SMPS data as suggested. However, no robust relationship was found between aerosol mass and the concentration of $pNO_3$. We were not able to do the same analysis as that in Neuman et al. (2003) due to poor resolution and missing data points of $HNO_3$ and the lack of $NH_3$ data.

*Q5: I cannot find the mentioned UHSAS data or DOAS data in the project archive. What does a "very good" agreement mean (line 142)? Again, quantifying the agreement and showing data would strengthen the paper.*
*Why is OH estimated using a prior study (line 246), when the OH measurements listed in Table 1 could be used?*

**Response:** No HONO data from DOAS is available in the project archive, because the ambient HONO concentrations ($11.2 \pm 4.3$ pptv in the PBL and $5.6 \pm 3.4$ in the FT as measured by LPAP) were mostly below the lower detection limits of the DOAS instrument (30 pptv) during the NOMADSS study. Good HONO measurements were made by both the DOAS ant the LPAP in wide power plant plumes during RF 7 over Ohio River Valley, and results have been intercompared (Extended Data Fig. 3 in Ye et al., 2016a). We found the HONO concentrations from the two instruments closely tracks each other, and the agreements were within the assessed uncertainties. The readers are encouraged to read the paper for more information.

Both this manuscript and the "prior" paper by Kaser et al. (2015) were based on results from the NOMADSS study, and the same OH measurement dataset was shared and used by the two papers. Since the information on OH levels during the flight had been published, it is appropriate to reference the paper.

*Q6: A very large photolysis rate for pNO3 is used to explain HONO formation, but this rate isn't consistent with the data shown. It is difficult for me to understand the difference between "determined photolysis rate" and "ambient photolysis rate" (section 3.3), but both are extremely large and comparable to the loss rate for isoprene. The nitrate photolysis rates give a nitrate lifetime of approx. one hour, which is less than the lifetime of NOx. How can nitrate ever accumulate in the atmosphere if its lifetime is so short? Are there any other studies that find a very short lifetime for nitrate? The large nitrate photolysis rate is inconsistent with the nitrate abundance and distribution reported here and cannot explain the HONO abundance.*

**Response:** Indeed, a very large photolysis rate constant was used for $pNO_3$ in our calculation. The $pNO_3$ photolysis rate constant was determined in the laboratory using the aerosol samples collected on board the C130 during the NOMADSS field study (Ye et al., 2017a). Several recent laboratory studies have demonstrated that surface nitric acid and particulate nitrate can be photolyzed at much higher rates than gaseous nitric acid, by 2-3 orders of magnitudes (Baergen and Donaldson, 2013; Du and Zhu, 2011; Ye et al., 2016b, 2017a; Zhou et al., 2003; Zhu et al., 2010, 2015). While $NO_2$ has been found to be the dominant product from $HNO_3$ photolysis on clean and dry laboratory surfaces (Ye et al., 2016b; Zhou et al., 2003; Zhu et al., 2010, 2015), HONO is the major product on natural surfaces and in ambient aerosols (Ye et al., 2016b, 2017a).

The "determined photolysis rate constant" ($J_{pNO_3}^N$) is the laboratory determined photolysis rate constant using the ambient aerosol samples. It has been normalized to tropical noontime conditions at ground level (solar zenith angle = 0 °), so that it can be compared with results in other studies. The $J_{pNO_3}^N$ value varies over a wide range, from $8.3 \times 10^{-5}$ s$^{-1}$ to $3.1 \times 10^{-4}$ s$^{-1}$ among the samples, with a median of $2.0 \times 10^{-4}$ s$^{-1}$ and a mean ($\pm 1$ standard deviation) of $1.9 (\pm 1.2) \times 10^{-4}$ s$^{-1}$. A median $J_{pNO_3}^N$ value of $2.0 \times 10^{-4}$ s$^{-1}$ was used in the calculation.

The "ambient photolysis rate constant" is the $pNO_3$ photolysis rate constant ($J_{pNO3}$) under the ambient conditions. It varies with the time of the day, the location, and the cloud coverage. $J_{pNO3}$ was calculated by scaling $J_{pNO_3}^N$ (~$2.0 \times 10^{-4}$ s$^{-1}$) to ambient light conditions using the measurement-derived $J_{HNO3}$ (Eq. 3).

Yes, some recent studies also showed the short lifetime of particulate nitrate in low-$NO_x$ environments (Reed et al., 2017; Ye et al., 2016a, 2017b). Many laboratory studies have also shown fast photolysis rate constant for surface $HNO_3$ and $pNO_3$ (Baergen and Donaldson, 2013, Ye et al., 2016b, 2017a; Zhou et al., 2003; Zhu et al, 2010; Zhu et al., 2015), lending support to our argument that $pNO_3$ photolysis can be an effective renoxification pathway. However, particulate nitrate is in a dynamic equilibrium with

gas-phase $HNO_3$, the later accounts for a larger (or even dominant) fraction of total nitrate ($pNO_3+HNO_3$) and is photochemically inert. The overall photolysis of $pNO_3+HNO_3$ would be much slower than indicated by $J_{pNO3}$. In addition, oxidation of $NO_x$ via several reactions will replenish the $pNO_3+HNO_3$ reservoir. Our results reported in this manuscript and in an earlier paper (Ye et al., 2016a) suggest that there is a rapid cycling in reactive nitrogen species in the low-$NO_x$ atmosphere, sustaining the observed levels of HONO and $pNO_3$.

*Q7: The different air mass types are not explained, and it isn't clear if or how they were separated. Benzene is used to identify urban plumes, but how are power plant plumes and biomass burning plumes identified? Could there be a large biomass burning plume contribution to the observations? Were some plumes a combination of sources? CO or acetonitrile measurements could be used to identify air mass influences. Similarly, SO2 was measured and could be used to identify power plant plumes. I could find no mention of any meteorological conditions. Without a more thorough description of the ambient conditions and ancillary measurements, it is very difficult to compare these results with other studies.*

**Response:** As suggested, we have added CO, acetonitrile and $SO_2$ as tracers to identify plumes in the revised manuscript (Figure S1). Based on the low levels of acetonitrile during the reported flights in this manuscript, we did not observe any significant contribution from biomass burning (Figure S1). The original assignments of plumes are further confirmed by these tracers: The CO peaks in plumes U1,U2 and U3 (A, B, C in the original Figures 2 and 7) suggest that they were under influenced by urban activities, and the lack of CO peaks in plumes P1-P4 (D, E, F, G in the original Figures 2 and 7) suggest that they were power plant plumes. A high $SO_2$ peak also accompanied a high $NO_x$ peak in the power plant plume P4 (G in the original Figures 2 and 7).

We did employ the meteorological information in our discussions, for examples, using the wind speed and wind direction to calculate the transport time of plumes from a power plant (original lines 358-362) and back trajectories in explaining horizontal HONO variations (original lines 401-409). Ancillary measurements, including OH, NO, $NO_2$, aerosol number and size distribution, isoprene, *J* values, …, were used in calculations and discussion throughout the manuscript.

*Q8: The large reduction in PBL mixing time (line 262) between noon and afternoon is very surprising and differs from previous studies. By noon in the summer, the mixing time should be much less than 1.5 h.*

**Response:** The mixing time can be influenced by several factors, such as the surface albedo and cloud coverage. The longer than expected mixing time was calculated using the vertical isoprene profile and may be due to the combined effect of these factors. We have also found significant variations in mixing time in RF #4, #5 and #17. Nevertheless, HONO photolytic lifetime was still much shorter than the mixing time even if the later was reduced by half; the conclusion of this section remains unchanged, i.e., the contribution of ground HONO source was not important to the overall HONO budget in the PBL, due to low ground source strength and/or slower transport than its photolysis loss.

*Q9: Relevant literature is not referenced, and the differences with previous measurements are not discussed. We published a very similar paper, using aircraft HONO measurements at the same time and location and under the same SAS umbrella (Neuman at al., HONO emission and production determined from airborne measurements over the Southeast U.S., JGR, 2016), but oddly, that paper is not referenced. We found that known HONO production mechanisms explained the HONO abundance, and we did not need to invoke unknown sources. In contrast, the studies referenced in the introduction (lines 29-30, line 103) report much larger values ranging from 100s of pptv to ppb levels. Why do the HONO values reported here differ from previous measurements, which range from indistinguishable from zero to ppbv levels? Meaningful comparisons to previous studies (some conducted at the same time and location) are essential for understanding the findings reported here.*

**Response:** We finished our first draft of this manuscript over two years ago, before the publication of the mentioned paper (Neuman et al., 2016).   Although we have made significant changes to the draft during the subsequent revisions, we failed to update the references.   We regret the omission. The paper by Neuman et al. (2016) has been referenced and discussed in the revised manuscript (lines 64, 95, 211, 215, 239, 282, 403).

We agree that meaningful comparisons to previous studies are essential for understanding the reported data.   We compared our results with those from other two airborne studies (Zhang et al., 2009; Li et al., 2014) in the original manuscript, and have added more discussions and comparisons with Neuman et al. (2016) in the revised manuscript.

Although the two aircraft studies, SENEX on NOAA's WP-3D and NOMADSS on NSF/NCAR's C-130, were conducted at the same time and location and under the same SAS umbrella, they had been focused on somewhat different objectives. The main objective of TROPHONO project (one of the three projects in NOMADSS) was to investigate daytime HONO formation mechanisms and the role of nitrate photolysis in aerosol particles in the cycling of reactive nitrogen species in the troposphere. All the C-130 flights were conducted in the daytime during NOMADSS except the RF#18 (from late-afternoon to midnight). And the results reported in this manuscript were mostly from rural background air masses, with only a few small urban and power plant plumes in RE#11.   On the other hand, the WP-3D spent far more time in various plumes and at nights during the SENEX (Neuman et al., 2016).   We would like to point out that there are actually no major disagreements between the two aircraft-based studies when the overlapped measurements are compared.   Similar to what reported by Neuman et al. (2016), we found that the $NO_x$-related reactions (mainly the $NO+OH$ reaction) accounted for nearly all the required HONO source in the large fresh power plant plume ($NO_x \sim 20$ ppbv) encountered during the RF #7 to Ohio River Valley (lines 375-378 in the original manuscript).   In the low-$NO_x$ background air masses, the mean HONO concentration was $11.2 \pm 4.3$ pptv in the PBL and $5.6 \pm 3.4$ in the free troposphere (Table 2), which is within the range from -15 pptv to 10 pptv ($\pm 15$ pptv uncertainty) (Neuman et al., 2016).

As pointed out by Andy Neuman, the studies referenced in Section "1. Introduction" reported significantly higher HONO concentrations, up to hundreds of pptv in the rural environments and several ppbv in the urban environments. The reported values include lower daytime and higher nighttime HONO concentrations. Most of these measurements were made on ground stations, and thus under direct influence of the ground HONO sources, including

direct emissions (combustion sources and soil emission sources), heterogeneous and photochemical reactions of precursors (e.g., $NO_2$, PAN and $HNO_3$) on ground surfaces, and gaseous reactions with elevated reactant concentrations. The measurements on aircrafts, on the other hand, minimized the influence of the ground HONO sources, as we discussed on section 3.2. Therefore, the airborne measurement data would provide a better insight into the HONO chemistry within an air parcel.

*Q10: Line 85 states that nearly all HONO measurements have been made at ground sites, but that dismisses the many studies of HONO vertical gradients using DOAS and from towers (e.g. Young et al, Vertically Resolved Measurements of Nighttime Radical Reservoirs in Los Angeles and Their Contribution to the Urban Radical Budget, ES&T, 2012; Stutz et al., Simultaneous DOAS and mist-chamber IC measurements of HONO in Houston, TX, Atmospheric Environment, 2010; Vandenboer 2013 in the references). And the authors themselves have many papers that detail airborne measurements.*

**Response:** We would like to point out that the HONO gradient measurements using DOAS and from towers are still ground-based, and that we did reference many of the related literatures (Kleffmann et al., 2003; Li et al., 2014; Stutz et al., 2002; Villena et al., 2011; Wong et al., 2011, 2012, 2013; Ye et al., 2015; Zhang et al., 2009) when discussing HONO vertical measurements and airborne measurements in the introduction (the paragraph starting line 85 in the original manuscript).

*Q11: smaller points I don't know what an N(V) level is (line 216)*

**Response:** We have much few data points of $HNO_3$, due to poor time resolution and more technical difficulties with the system (bubble formation/baseline shift, especially at high altitudes). The $HNO_3$ levels in the PBL were $305 \pm 87$ pptv in RF4, $291 \pm 81$ pptv in RF 5, $342 \pm 108$ pptv in RF11, $105 \pm 38$ pptv in RF 17, and $206 \pm 73$ pptv in RF18, accounting for 70% - 85% of (N(V).

*Q12: Data averaging is not explained. The time resolution of HONO and pNO3 are listed as 3 min and 6 min, yet 1 min data are shown. How are the data averaged in figure 3? The values do not match those shown in Figure 2, but the binning and averaging are never described.*

**Response:** The absorbance signals were sampled at 1 Hz, much higher rate than the time resolutions of HONO and $pNO_3$ (see Figure 1 in this Response). The 1-min or 3-min averages were used to convert absorbance signals to concentrations, based on flow rate and calibration information (please see our responses to Q1 and Q2 for more details on method and data processing). We have added the above information to the revised manuscript and have used 3-min HONO data and 6-min $pNO_3$ data for the revised figures, as suggested.

*Q14: Figure 2 shows pNO3 in ppbv, which is in error.*

**Response:** Thank you for pointing out the error; the error has been corrected.

**References**

Baergen, A. M., and Donaldson, D. J.: Photochemical renoxification of nitric acid on real urban grime, Environ. Sci. Technol., 47, 815-820, 10.1021/es3037862, 2013.

Du, J., and Zhu, L.: Quantification of the absorption cross sections of surface-adsorbed nitric acid in the 335-365 nm region by Brewster angle cavity ring-down spectroscopy, Chem. Phys. Lett., 511, 213-218, 10.1016/j.cplett.2011.06.062, 2011.

Heald, C. L., et al.: Atmospheric ammonia and particulate inorganic nitrogen over the United States, Atmos. Chem. Phys., 12, 10295–10312, 2012.

Kaser, L., et al.: chemistry-turbulence interactions and mesoscale variability influence the cleansing efficiency of the atmosphere, Geophys. Res. Lett., 42, doi:10.1002/2015GL066641, 2015.

Kleffmann, J., Kurtenbach, R., Lorzer, J., Wiesen, P., Kalthoff, N., Vogel, B., and Vogel, V.: Measured and simulated vertical profiles of nitrous acid - Part I: Field measurements, Atmos. Environ., 37, 2949-2955, 2003.

Li, X., et al.: Missing gas-phase source of HONO inferred from Zeppelin measurements in the troposphere, Science, 344, 292-296, 2014.

Neuman, J. A., et al.: Variability in ammonium nitrate formation and nitric acid depletion with altitude and location over California, J. Geophys. Res., 108(D17), 4557, doi:10.1029/2003JD003616, 2003.

Neuman, J. A., et al., HONO emission and production determined from airborne measurements over the Southeast U.S., J. Geophys. Res. Atmos., 121, 9237–9250, doi:10.1002/2016JD025197, 2016

Reed, C. et al.: Evidence for renoxification in the tropical marine boundary layer, Atmos. Chem. Phys., 17, 4081–4092, 2017.

Stutz, J., Alicke, B., and Neftel, A.: Nitrous acid formation in the urban atmosphere: Gradient measurements of NO2 and HONO over grass in Milan, Italy, J. Geophys. Res.-Atmos., 107, Artn 8192,10.1029/2001jd000390, 2002. Wong, K. W., Oh, H. J., Lefer, B. L., Rappengluck, B., and Stutz, J.: Vertical profiles of nitrous acid in the nocturnal urban atmosphere of Houston, TX, Atmos. Chem. Phys., 11, 3595-3609, 2011.

Villena, G., Kleffmann, J., Kurtenbach, R., Wiesen, P., Lissi, E., Rubio, M. A., Croxatto, G., and Rappengluck, B.: Vertical gradients of HONO, NOx and O3 in Santiago de Chile, Atmos. Environ., 45, 3867-3873, 2011.

Wong, K. W., Tsai, C., Lefer, B., Haman, C., Grossberg, N., Brune, W. H., Ren, X., Luke, W., and Stutz, J.: Daytime HONO vertical gradients during SHARP 2009 in Houston, TX, Atmos. Chem. Phys., 12, 635-652, 2012.

Wong, K. W., Tsai, C., Lefer, B., Grossberg, N., and Stutz, J.: Modeling of daytime HONO vertical gradients during SHARP 2009, Atmos. Chem. Phys., 13, 3587-3601, 2013.

Ye, C. X., Zhou, X. L., Pu, D., Stutz, J., Festa, J., Spolaor, M., Cantrell, C., Mauldin, R. L., Weinheimer, A., and Haggerty, J.: Comment on "Missing gas-phase source of HONO inferred from Zeppelin measurements in the troposphere", Science, 348, 10.1126/science.aaa1992, 2015.

Ye, C., et al.: Rapid cycling of reactive nitrogen in the marine boundary layer, Nature, 532, 489-491, 2016a

Ye, C., Gao, H., Zhang, N., and Zhou, X.: Photolysis of nitric Acid and nitrate on natural and

artificial surfaces, Environ. Sci. Technol., 50, 3530-3536, 2016b.

Ye, C., Zhang, N., Gao, H., and Zhou, X.: Photolysis of particulate nitrate as a source of HONO and $NO_x$, Environ. Sci. Technol., DOI: 10.1021/acs.est.7b00387, 2017a.

Ye, C., Heard, D.E., and Whalley, L.K.: Evaluation of novel routes for NOx formation in remote regions, Environ. Sci. Technol., DOI: 10.1021/acs.est.6b06441, 2017b.

Zhang, N., Zhou, X., Shepson, P. B., Gao, H., Alaghmand, M., and Stirm, B.: Aircraft measurement of HONO vertical profiles over a forested region, Geophys. Res. Lett., 36, L15820,10.1029/2009gl038999, 2009.

Zhang, N., Zhou, X., Bertman, S., Tang, D., Alaghmand, M., Shepson, P. B., and Carroll, M. A.: Measurements of ambient HONO concentrations and vertical HONO flux above a northern Michigan forest canopy, Atmos. Chem. Phys., 12, 8285-8296, 2012.

Zhou, X., Gao, H., He, Y., Huang, G., Bertman, S. B., Civerolo, K., and Schwab, J.: Nitric acid photolysis on surfaces in low-NOx environments: Significant atmospheric implications, Geophys. Res. Lett., 30, 2217, 10.1029/2003gl018620, 2003.

Zhu, C., Xiang, B.. Chu, L.T., and Zhu, L.: 308 nm Photolysis of nitric acid in the gas phase, on aluminum surfaces, and on ice films, J. Phys. Chem. A, 114, 2561-2568, 2010.

Zhu, L., Sangwan, M., Huang, L., Du, J., and Chu, L.T.: Photolysis of nitric acid at 308 nm in the absence and in the presence of water vapor, J. Phys. Chem. A 2015, 119, 4907-4914, 2015.

---

## Author Comment (AC2) · 3 May 2018

Response to interactive comment on manuscript on "Tropospheric HONO Distribution and Chemistry in the Southeast U.S." by D. Parrish

C. Ye (c.ye@pku.edu.cn) and X. Zhou (xianliang.zhou@health.ny.gov)

We completely agree with David Parrish that "Extraordinary claims require extraordinary evidence." However, we disagree that our finding that ". . .the sum of all known NOx-related HONO formation mechanisms was found to account for less 20% of the daytime HONO source in the background terrestrial air masses, . . .." is an extraordinary claim. In high NOx environments, such as urban atmosphere and power plant and biomass burning plumes, NOx is known to be the dominant precursor to HONO. How-

ever, in low NOx environments, such as the rural regions in the Southeast US, other precursors become more important. In fact, there have been many reports in literature, based on both field and laboratory results, demonstrating that several processes other than reactions involving ambient NOx can lead to the production of HONO. Nitrate photolysis in snowpack has been found to be a major source for HONO and NOx during the polar spring and summer in the polar regions (Beine et al., 2002, 2008; Honrath et al., 2000, 2002; Zhou et al., 2001). In low-NOx rural and forested regions, photolysis of nitric acid on the forest canopy surface has been found to be the major daytime HONO source (Ye et al., 2016a; Zhou et al., 2002, 2003, 2011). Photolysis of particulate nitrate has been found to be the major HONO source in the low-NOx marine boundary layer (Reed et al, 2017; Ye et al., 2016b). And in agricultural regions, biochemical process in the soils (denitrification or nitrification) has been found to account for the majority of HONO budget (Oswald et al., 2013; Su et al., 2012; Meusel et al., 2018).

We estimated the HONO formation rates from known homogeneous and heterogeneous NOx reactions, with a suit of parameters measured on board the C-130, and found the sum of these mechanisms to contribute less than 20% of the total HONO source strength in the background air masses. Most of the remaining so-called "unknown" 80% can actually be accounted for by the photolysis of particulate nitrate (lines 302 - 331 in the original manuscript). This finding is consistent with several reported laboratory studies that the photolysis of surface nitric acid and particulate nitrate is 2 - 3 orders faster than that of gaseous nitric acid (Baergen and Donaldson, 2013, Ye et al., 2016a, 2017a; Zhou et al., 2003; Zhu et al, 2010, 2015), producing mostly $NO_2$ on clean dry surface (Ye et al., 2016a; Zhou et al., 2003; Zhu et al, 2010) and mainly HONO on natural surfaces and ambient aerosols (Ye et al., 2016a, 2017a).

We would also like to point out that while HONO photolysis can be a significant or even a major HOx source on the ground level in both rural and urban atmosphere (Acker et al., 2006a,b; Elshorbany et al., 2010; Kleffmann et al., 2003, 2005; Villena et al., 2011), it was found unimportant compared to photolyses of $O_3$ and HCHO in the background

air masses aloft over the Southeast US. At the observed levels of 5-11 pptv, the answer to the HONO source question is unlikely to significantly affect our understanding of HOx chemistry in the rural troposphere. On the other hand, since HONO was found to be mainly produced from photolysis of particulate nitrate, it is an important intermediate product of a photochemical renoxification process recycling nitric acid and nitrate back to NOx.

We regret that we did not reference the recent paper by Neuman et al. (2016). We prepared and finished our first draft of this manuscript over two years ago, before the publication of the mentioned paper. Although we have made significant changes to the first draft during the subsequent revisions, we failed to update the references. We have referenced and discussed the paper in the revised manuscript (lines 64, 95, 211, 215, 239, 282, 403). It is important to point out that there is no major disagreement in the results between the two aircraft-based studies. Similar to what reported by Neuman et al. (2016), we found that the NOx-related reactions (mainly NO+OH reaction) accounted for nearly all the required HONO source in the large fresh power plant plume (NOx $\sim$ 20 ppbv) encountered during the RF #7 to Ohio River Valley (lines 375-378 in the original manuscript). In the smaller and more diluted power plant plume G in the original Figures 2c and 7b (NOx $\sim$1.8 ppb), NOx-related reactions contribute to a major fraction (52%) of the total required HONO source (the original Figure 7b). In the low-NOx background air masses, the mean HONO concentration was 11.2 $\pm$ 4.3 pptv in the PBL and 5.6 $\pm$ 3.4 in the free troposphere (Table 2), which is again in agreement the value reported by Neuman et al (2016) "indistinguishable from zero within the 15 parts per trillion by volume measurement uncertainty." We would further argue that while the CIMS instrument, with detection limits of 40 pptv for 1-s data and 15 pptv for 30-min averaging, is capable of producing high quality data in the plumes, it does not have the sensitivity to measure low levels of HONO in the low-NOx background atmosphere. The conclusion based on its below-detection-limit measurements and on the extrapolations from combustion plumes to low-NOx background atmosphere is not reliable and thus should not be used to rule out the findings based on our measurement

in the low-NOx rural atmosphere. The relative contribution of NOx-related reactions is in the order of power plant plume (NOx $\sim$ 1- 20 ppb) > urban plume (NOx $\sim$ 1 ppb) > background terrestrial air masses (NOx $\sim$100-300 pptv). That is, the relative contribution from NOx-related reactions to the required HONO source is highly dependent on the NOx regimes. While the conclusion we draw in the high NOx regime in large power plant plumes is not different from that by Neuman et al. (2016), our measurements have added new and valuable HONO budget information in low NOx regime to the literature.

We appreciate the question regarding potential problems with experimental design/measurement technique. More detailed descriptions and discussions on HONO measurement technique and set up have been provided in our response to Andy Neuman's comment (#1 and #2). The wet chemistry-based techniques, including the LPAP used in this study, can provide exceptionally high sensitivity for HONO. However, the measurements by these techniques have been treated with caution and suspicion due to potential interferences from ambient constituents. We have made major and continued efforts in the past two decades to minimize and correct for the potential interferences. For examples, we found that shielding the inlet line from sunlight could prevent photochemical formation of HONO on the inlet wall surface (Zhou et al., 2002b). Results from many field and laboratory tests we conducted so far have indicated that heating the inlet line can effectively minimize the HONO loss to and/or HONO formation from heterogeneous NO2 reactions on inlet wall surface (see Figure 3 in the response to Andy Neuman's comment). We have used Na2CO3-coated denuder to generate "zero-HONO" air by selectively removing HONO (and acidic species) from ambient air to established measurement baselines. The subtraction of "zero-HONO" air baselines from ambient signals effectively eliminate the potential interference from HONO precursors, such as NOx, PAN and particulate nitrite (Zhang et al., 2012; Figures 1 and 2 in the Response to A.Neuman's Comment). To check the effectiveness of our background correction procedure and to validate the LPAS technique, we have compared the HONO concentrations measured by the LPAS and by a limb-scanning differential optical absorption spectroscopy (DOAS) instruments on board the C-130 in large power plant plumes during the NOMADSS campaign, and found very good agreement between the two measurements (Ye et al., 2016b). Therefore, we have high confident with our HONO data measured on the C-130 during the NOMADSS field study, and we stand by our findings that the photolysis of particulate nitrate is the major daytime HONO source and NOx-related reactions is an only minor HONO contributor in the low-NOx TBL over Southeast U.S.

References

Acker, K., Moller, D., Wieprecht, W., Meixner, F. X., Bohn, B., Gilge, S., Plass-Dulmer, C., and Berresheim, H.: Strong daytime production of OH from HNO2 at a rural mountain site, Geophys. Res. Lett., 33, L02809,10.1029/2005gl024643, 2006.

Acker, K., et al.: Nitrous acid in the urban area of Rome, Atmos. Environ., 40, 3123-3133, 2006b. Baergen, A. M., and Donaldson, D. J.: Photochemical renoxification of nitrica on real urban grime, Environ. Sci. Technol., 47, 815-820, 10.1021/es3037862, 2013.

Beine, H., Domine, F., Simpson, W. Honrath, R.E., Sparapani, R., Zhou, X., and King, M.: Snow-pile and chamber experiments during the Polar Sunrise Experiment 'Alert 2000': exploration of nitrogen chemistry. Atmos. Environ. 2002, 36, 2707-2719, 2002.

Beine, H., Colussi, A.J., Amoroso, A., Esposito, G., Montagnoli, M., and Hoffmann, M.R.: HONO emissions from snow surfaces, Environ. Res. Lett., 3, 045005, 2008.

Elshorbany, Y. F., Kleffmann, J., Kurtenbach, R., Lissi, E., Rubio, M., Villena, G., Gramsch, E., Rickard, A. R., Pilling, M. J., and Wiesen, P.: Seasonal dependence of the oxidation capacity of the city of Santiago de Chile, Atmos. Environ., 44, 5383-5394, 10.1016/j.atmosenv.2009.08.036, 2010.

Honrath, R. E., Peterson, M. C., Dziobak, M. P., Dibb, J. E., Arsenault, M. A., and Green S. A.: Release of NOx from sunlight-irradiated midlatitude snow, Geophys. Res. Lett.,

Interactive
comment

27, 2237-2240, 2000.

Honrath R. E., Lu, Y., Peterson, M.C., Dibb, J.E., Arsenault, M.A., Cullen, N.J., and Steffen, K.: Vertical fluxes of NOx, HONO, and HNO3 above the snowpack at Summit, Greenland Atmos. Environ., 36 2629-40, 2002.

Kleffmann, J., Kurtenbach, R., Lorzer, J., Wiesen, P., Kalthoff, N., Vogel, B., and Vogel, V.: Measured and simulated vertical profiles of nitrous acid - Part I: Field measurements, Atmos. Environ., 37, 2949-2955, 2003.

Kleffmann, J., et al.: Daytime formation of nitrous acid: A major source of OH radicals in a forest, Geophys. Res. Lett., 32, doi:10.1029/2005GL022524, 2005.

Meusel, H. et al.: Emission of nitrous acid from soil and biological soil crusts represents an important source of HONO in the remote atmosphere in Cyprus, Atmos. Chem. Phys., 18, 799–813, 2018.

Neuman, J. A., et al.: HONO emission and production determined from airborne measurements over the Southeast U.S., J. Geophys. Res. Atmos., 121, 9237–9250, doi:10.1002/2016JD025197, 2016. Oswald, R., et al.: HONO emissions from soil bacteria as a major source of atmospheric reactive nitrogen, Science, 341, 1233-1235, DOI: 10.1126/science.1242266, 2013.

Reed, C. et al.: Evidence for renoxification in the tropical marine boundary layer, Atmos. Chem. Phys., 17, 4081–4092, 2017.

Ren, X., et al.: OH and HO2 chemistry in the urban atmosphere of New York City. Atmos. Environ. 37, 3639-3651, 2003.

Su, H., Cheng, Y. F., Oswald, R., Behrendt, T., Trebs, I., Meixner, F. X., Andreae, M. O., Cheng, P., Zhang, Y., and Poschl, U.: Soil Nitrite as a Source of Atmospheric HONO and OH Radicals, Science, 333, 1616-1618, 10.1126/science.1207687, 2011.

Villena, G., et al.: Vertical gradients of HONO, NOx and O3 in Santiago de Chile,

Atmos. Environ., 45, 3867-3873, 2011.

Ye, C., Gao, H., Zhang, N., and Zhou, X.: Photolysis of nitric Acid and nitrate on natural and artificial surfaces, Environ. Sci. Technol., 50, 3530-3536, 2016a.

Ye, C., et al.: Rapid cycling of reactive nitrogen in the marine boundary layer, Nature, 532, 489-491, 2016b.

Ye, C., Zhang, N., Gao, H., and Zhou, X.: Photolysis of particulate nitrate as a source of HONO and NOx, Environ. Sci. Technol., DOI: 10.1021/acs.est.7b00387, 2017a.

Ye, C., Heard, D.E., and Whalley, L.K.: Evaluation of novel routes for NOx formation in remote regions, Environ. Sci. Technol., DOI: 10.1021/acs.est.6b06441, 2017b.

Zhou, X., H. J. Beine, H.J., Honrath, R.E., Fuentes, J.D., Simpson, W., Shepson, P.B., and J. W. Bottenheim, J.W.: Snowpack photochemical production as a source for HONO in the Arctic boundary layer in spring time, Geophys. Res. Lett, 28:4087-4090, 2001.

Zhou, X., Civerolo, K., Dai, H., Huang, G., Schwab, J., and Demerjian, K.: Summertime nitrous acid chemistry in the atmospheric boundary layer at a rural site in New York State, J. Geophys. Res., 107, doi:10.1029/2001JD001539, 2002a.

Zhou, X., He, Y.,Huang, G.,Thornberry, T.D.,. Carroll, M.A., and Bertman, S.B.: Photo-chemical production of HONO on glass sample manifold wall surface, Geophys. Res. Lett, 29, doi:10.1029/2002GL015080, 2002b.

Zhou, X., Gao, H., He, Y., Huang, G., Bertman, S. B., Civerolo, K., and Schwab, J.: Nitric acid photolysis on surfaces in low-NOx environments: Significant atmospheric implications, Geophys. Res. Lett., 30, 2217, 10.1029/2003gl018620, 2003.

Zhou, X., G. Huang, G., Civerolo, K., Roychowdhury, U., and Demerjian, K.L.: Summertime observations of HONO, HCHO, and O3 at the summit of Whiteface Mountain, New York, J. Geophys. Res., 112, doi:10.1029/2006JD007256, 2007.

Zhou, X., Zhang, N., TerAvest, M., Tang, D., Hou, J., Bertman, S., Alaghmand, M., Shepson, P. B., Carroll, M. A., Griffith, S., Dusanter, S., and Stevens, P. S.: Nitric acid photolysis on forest canopy surface as a source for tropospheric nitrous acid, Nature Geosci., 4, 440-443, 10.1038/NGEO1164, 2011.

Zhu, C., Xiang, B.. Chu, L.T., and Zhu, L.: 308 nm Photolysis of nitric acid in the gas phase, on aluminum surfaces, and on ice films, J. Phys. Chem. A, 114, 2561-2568, 2010.

Zhu, L., Sangwan, M., Huang, L., Du, J., and Chu, L.T.: Photolysis of nitric acid at 308 nm in the absence and in the presence of water vapor, J. Phys. Chem. A 2015, 119, 4907-4914, 2015.

Please also note the supplement to this comment:
https://www.atmos-chem-phys-discuss.net/acp-2018-105/acp-2018-105-AC2-supplement.pdf

[Figure]

**Supplement:**

C. Ye (c.ye@pku.edu.cn) and X. Zhou (xianliang.zhou@health.ny.gov)

*General comments*

*"Extraordinary claims require extraordinary evidence" is a phrase made popular by Carl Sagan [Rational Wiki]. This is particularly relevant to the Ye et al. [2018] paper. The authors make the extraordinary claim that ". . .the sum of all known NOx-related HONO formation mechanisms was found to account for less 20% of the daytime HONO source in the background terrestrial air masses, . . ..". If this claim were true, it would possibly force a reassessment of our understanding of HOx and NOx budgets of the troposphere, depending on the details of the other 80% of the (unknown) sources. However, the evidence presented Ye et al., [2018] does not justify that claim. In fact, Neuman et al. [2016] conclude: "Outside of recently emitted plumes from known combustion sources, HONO mixing ratios measured several hundred meters above ground level were indistinguishable from zero within the 15 parts per trillion by volume measurement uncertainty. The results reported here do not support the existence of a ubiquitous unknown HONO source that produces significant HONO concentrations in the lower troposphere." The conclusion of these two studies disagree strongly, yet the reported measurements were made from different aircraft with different instrumentation, but in the same region of the country over the same time period, summer 2013. Unfortunately, Ye et al., [2018] do not discuss or even cite Neuman et al. [2016]. The differences in the results reported in these two papers point to clear experimental problems in the measurements. Until these problems are resolved, the extraordinary claim of Ye et al. [2018] should not be published.*

*References*

*Neuman, J. A., et al. (2016), HONO emission and production determined from airborne measurements over the Southeast U.S., J. Geophys. Res. Atmos., 121, 9237–9250, doi:10.1002/2016JD025197.*

*Rational Wiki, ttps://rationalwiki.org/wiki/Extraordinary_claims_require_extraordinary_evidenceh*

*Ye, C., et al. (2018), Tropospheric HONO distribution and chemistry in the Southeast U.S., Atmos. Chem. Phys. Discuss., https://doi.org/10.5194/acp-2018-105.*

**Response:** We completely agree with David Parrish that "Extraordinary claims require extraordinary evidence." However, we disagree that our finding that ". . .the sum of all known $NO_x$-related HONO formation mechanisms was found to account for less 20% of the daytime HONO source in the background terrestrial air masses, . . .." is an extraordinary claim. In high $NO_x$ environments, such as urban atmosphere and power plant and biomass burning plumes, $NO_x$ is known to be the dominant precursor to HONO. However, in low $NO_x$ environments, such as the rural regions in the Southeast US, other precursors become more important. In fact, there have been many reports in literature, based on both field and laboratory results, demonstrating that several processes other than reactions involving ambient $NO_x$ can lead to the production of HONO. Nitrate photolysis in snowpack has been found to be a major source

for HONO and $NO_x$ during the polar spring and summer in the polar regions (Beine et al., 2002, 2008; Honrath et al., 2000, 2002; Zhou et al., 2001). In low-$NO_x$ rural and forested regions, photolysis of nitric acid on the forest canopy surface has been found to be the major daytime HONO source (Ye et al., 2016a; Zhou et al., 2002, 2003, 2011). Photolysis of particulate nitrate has been found to be the major HONO source in the low-$NO_x$ marine boundary layer (Reed et al, 2017; Ye et al., 2016b). And in agricultural regions, biochemical process in the soils (denitrification or nitrification) has been found to account for the majority of HONO budget (Oswald et al., 2013; Su et al., 2012; Meusel et al., 2018).

We estimated the HONO formation rates from known homogeneous and heterogeneous $NO_x$ reactions, with a suit of parameters measured on board the C-130, and found the sum of these mechanisms to contribute less than 20% of the total HONO source strength in the background air masses. Most of the remaining so-called "unknown" 80% can actually be accounted for by the photolysis of particulate nitrate (lines 302 - 331 in the original manuscript). This finding is consistent with several reported laboratory studies that the photolysis of surface nitric acid and particulate nitrate is 2 - 3 orders faster than that of gaseous nitric acid (Baergen and Donaldson, 2013, Ye et al., 2016a, 2017a; Zhou et al., 2003; Zhu et al, 2010, 2015), producing mostly $NO_2$ on clean dry surface (Ye et al., 2016a; Zhou et al., 2003; Zhu et al, 2010) and mainly HONO on natural surfaces and ambient aerosols (Ye et al., 2016a, 2017a).

We would also like to point out that while HONO photolysis can be a significant or even a major $HO_x$ source on the ground level in both rural and urban atmosphere (Acker et al., 2006a,b; Elshorbany et al., 2010; Kleffmann et al., 2003, 2005; Villena et al., 2011), it was found unimportant compared to photolyses of $O_3$ and HCHO in the background air masses **aloft** over the Southeast US. At the observed levels of 5-11 pptv, the answer to the HONO source question is unlikely to significantly affect our understanding of $HO_x$ chemistry in the rural troposphere.   On the other hand, since HONO was found to be mainly produced from photolysis of particulate nitrate, it is an important intermediate product of a photochemical renoxification process recycling nitric acid and nitrate back to $NO_x$.

We regret that we did not reference the recent paper by Neuman et al. (2016). We prepared and finished our first draft of this manuscript over two years ago, before the publication of the mentioned paper. Although we have made significant changes to the first draft during the subsequent revisions, we failed to update the references.   We have referenced and discussed the paper in the revised manuscript (lines 64, 95, 211, 215, 239, 282, 403).

It is important to point out that there is no major disagreement in the results between the two aircraft-based studies. Similar to what reported by Neuman et al. (2016), we found that the $NO_x$-related reactions (mainly NO+OH reaction) accounted for nearly all the required HONO source in the large fresh power plant plume ($NO_x$ ~ 20 ppbv) encountered during the RF #7 to Ohio River Valley (lines 375-378 in the original manuscript). In the smaller and more diluted power plant plume G in the original Figures 2c and 7b ($NO_x$ ~1.8 ppb), $NO_x$-related reactions contribute to a major fraction (52%) of the total required HONO source (the original Figure 7b).   In the low-$NO_x$ background air masses, the mean HONO concentration was 11.2 ± 4.3 pptv in the PBL and 5.6 ± 3.4 in the free troposphere (Table 2), which is again in agreement the value reported by Neuman et al (2016) "indistinguishable from zero **within the 15 parts per trillion by volume measurement uncertainty**."

We would further argue that while the CIMS instrument, with detection limits of 40 pptv

for 1-s data and 15 pptv for 30-min averaging, is capable of producing high quality data in the plumes, it does not have the sensitivity to measure low levels of HONO in the low-$NO_x$ background atmosphere. The conclusion based on its below-detection-limit measurements and on the extrapolations from combustion plumes to low-$NO_x$ background atmosphere is not reliable and thus should not be used to rule out the findings based on our measurement in the low-$NO_x$ rural atmosphere. The relative contribution of $NO_x$-related reactions is in the order of power plant plume ($NO_x$ ~ 1- 20 ppb) > urban plume ($NO_x$ ~ 1 ppb) > background terrestrial air masses ($NO_x$ ~100-300 pptv). That is, the relative contribution from $NO_x$-related reactions to the required HONO source is highly dependent on the $NO_x$ regimes. While the conclusion we draw in the high $NO_x$ regime in large power plant plumes is not different from that by Neuman et al. (2016), our measurements have added new and valuable HONO budget information in low $NO_x$ regime to the literature.

We appreciate the question regarding potential problems with experimental design/measurement technique. More detailed descriptions and discussions on HONO measurement technique and set up have been provided in our response to Andy Neuman's comment (#1 and #2). The wet chemistry-based techniques, including the LPAP used in this study, can provide exceptionally high sensitivity for HONO. However, the measurements by these techniques have been treated with caution and suspicion due to potential interferences from ambient constituents. We have made major and continued efforts in the past two decades to minimize and correct for the potential interferences. For examples, we found that shielding the inlet line from sunlight could prevent photochemical formation of HONO on the inlet wall surface (Zhou et al., 2002b). Results from many field and laboratory tests we conducted so far have indicated that heating the inlet line can effectively minimize the HONO loss to and/or HONO formation from heterogeneous $NO_2$ reactions on inlet wall surface (see Figure 3 in the response to Andy Neuman's comment). We have used $Na_2CO_3$-coated denuder to generate "zero-HONO" air by selectively removing HONO (and acidic species) from ambient air to established measurement baselines. The subtraction of "zero-HONO" air baselines from ambient signals effectively eliminate the potential interference from HONO precursors, such as $NO_x$, PAN and particulate nitrite (Zhang et al., 2012; Figures 1 and 2 in the Response to A.Neuman's Comment). To check the effectiveness of our background correction procedure and to validate the LPAS technique, we have compared the HONO concentrations measured by the LPAS and by a limb-scanning differential optical absorption spectroscopy (DOAS) instruments on board the C-130 in large power plant plumes during the NOMADSS campaign, and found very good agreement between the two measurements (Ye et al., 2016b). Therefore, we have high confident with our HONO data measured on the C-130 during the NOMADSS field study, and we stand by our findings that the photolysis of particulate nitrate is the major daytime HONO source and $NO_x$-related reactions is an only minor HONO contributor in the low-$NO_x$ TBL over Southeast U.S.

**References**

Acker, K., Moller, D., Wieprecht, W., Meixner, F. X., Bohn, B., Gilge, S., Plass-Dulmer, C., and Berresheim, H.: Strong daytime production of OH from $HNO_2$ at a rural mountain site,

Geophys. Res. Lett., 33, L02809,10.1029/2005gl024643, 2006.

Acker, K., et al.: Nitrous acid in the urban area of Rome, Atmos. Environ., 40, 3123-3133, 2006b.

Baergen, A. M., and Donaldson, D. J.: Photochemical renoxification of nitrica on real urban grime, Environ. Sci. Technol., 47, 815-820, 10.1021/es3037862, 2013.

Beine, H., Domine, F., Simpson, W. Honrath, R.E., Sparapani, R., Zhou, X., and King, M.: Snow-pile and chamber experiments during the Polar Sunrise Experiment 'Alert 2000': exploration of nitrogen chemistry. Atmos. Environ. 2002, 36, 2707-2719, 2002.

Beine, H., Colussi, A.J., Amoroso, A., Esposito, G., Montagnoli, M., and Hoffmann, M.R.: HONO emissions from snow surfaces, Environ. Res. Lett., 3, 045005, 2008.

Elshorbany, Y. F., Kleffmann, J., Kurtenbach, R., Lissi, E., Rubio, M., Villena, G., Gramsch, E., Rickard, A. R., Pilling, M. J., and Wiesen, P.: Seasonal dependence of the oxidation capacity of the city of Santiago de Chile, Atmos. Environ., 44, 5383-5394, 10.1016/j.atmosenv.2009.08.036, 2010.

Honrath, R. E., Peterson, M. C., Dziobak, M. P., Dibb, J. E., Arsenault, M. A., and Green S. A.: Release of NOx from sunlight-irradiated midlatitude snow, Geophys. Res. Lett., 27, 2237-2240, 2000.

Honrath R. E., Lu, Y., Peterson, M.C., Dibb, J.E., Arsenault, M.A., Cullen, N.J., and Steffen, K.: Vertical fluxes of NOx, HONO, and HNO3 above the snowpack at Summit, Greenland Atmos. Environ., 36 2629-40, 2002.

Kleffmann, J., Kurtenbach, R., Lorzer, J., Wiesen, P., Kalthoff, N., Vogel, B., and Vogel, V.: Measured and simulated vertical profiles of nitrous acid - Part I: Field measurements, Atmos. Environ., 37, 2949-2955, 2003.

Kleffmann, J., et al.: Daytime formation of nitrous acid: A major source of OH radicals in a forest, Geophys. Res. Lett., 32, doi:10.1029/2005GL022524, 2005.

Meusel, H. et al.: Emission of nitrous acid from soil and biological soil crusts represents an important source of HONO in the remote atmosphere in Cyprus, Atmos. Chem. Phys., 18, 799–813, 2018.

Neuman, J. A., et al.: HONO emission and production determined from airborne measurements over the Southeast U.S., J. Geophys. Res. Atmos., 121, 9237–9250, doi:10.1002/2016JD025197, 2016.

Oswald, R., et al.: HONO emissions from soil bacteria as a major source of atmospheric reactive nitrogen, Science, 341, 1233-1235, DOI: 10.1126/science.1242266, 2013.

Reed, C. et al.: Evidence for renoxification in the tropical marine boundary layer, Atmos. Chem. Phys., 17, 4081–4092, 2017.

Ren, X., et al.: OH and $HO_2$ chemistry in the urban atmosphere of New York City. Atmos. Environ. 37, 3639-3651, 2003.

Su, H., Cheng, Y. F., Oswald, R., Behrendt, T., Trebs, I., Meixner, F. X., Andreae, M. O., Cheng, P., Zhang, Y., and Poschl, U.: Soil Nitrite as a Source of Atmospheric HONO and OH Radicals, Science, 333, 1616-1618, 10.1126/science.1207687, 2011.

Villena, G., et al.: Vertical gradients of HONO, $NO_x$ and $O_3$ in Santiago de Chile, Atmos. Environ., 45, 3867-3873, 2011.

Ye, C., Gao, H., Zhang, N., and Zhou, X.: Photolysis of nitric Acid and nitrate on natural and artificial surfaces, Environ. Sci. Technol., 50, 3530-3536, 2016a.

Ye, C., et al.: Rapid cycling of reactive nitrogen in the marine boundary layer, Nature, 532,

489-491, 2016b.

Ye, C., Zhang, N., Gao, H., and Zhou, X.: Photolysis of particulate nitrate as a source of HONO and $NO_x$, Environ. Sci. Technol., DOI: 10.1021/acs.est.7b00387, 2017a.

Ye, C., Heard, D.E., and Whalley, L.K.: Evaluation of novel routes for NOx formation in remote regions, Environ. Sci. Technol., DOI: 10.1021/acs.est.6b06441, 2017b.

Zhou, X., H. J. Beine, H.J., Honrath, R.E., Fuentes, J.D., Simpson, W., Shepson, P.B., and J. W. Bottenheim, J.W.: Snowpack photochemical production as a source for HONO in the Arctic boundary layer in spring time, Geophys. Res. Lett, 28:4087-4090, 2001.

Zhou, X., Civerolo, K., Dai, H., Huang, G., Schwab, J., and Demerjian, K.: Summertime nitrous acid chemistry in the atmospheric boundary layer at a rural site in New York State, *J. Geophys. Res.*, *107*, doi:10.1029/2001JD001539, 2002a.

Zhou, X., He, Y.,Huang, G.,Thornberry, T.D.,. Carroll, M.A., and Bertman, S.B.: Photochemical production of HONO on glass sample manifold wall surface, Geophys. Res. Lett, 29, doi:10.1029/2002GL015080, 2002b.

Zhou, X., Gao, H., He, Y., Huang, G., Bertman, S. B., Civerolo, K., and Schwab, J.: Nitric acid photolysis on surfaces in low-NOx environments: Significant atmospheric implications, Geophys. Res. Lett., 30, 2217, 10.1029/2003gl018620, 2003.

Zhou, X., G. Huang, G., Civerolo, K., Roychowdhury, U., and Demerjian, K.L.: Summertime observations of HONO, HCHO, and $O_3$ at the summit of Whiteface Mountain, New York, J. Geophys. Res., 112, doi:10.1029/2006JD007256, 2007.

Zhou, X., Zhang, N., TerAvest, M., Tang, D., Hou, J., Bertman, S., Alaghmand, M., Shepson, P. B., Carroll, M. A., Griffith, S., Dusanter, S., and Stevens, P. S.: Nitric acid photolysis on forest canopy surface as a source for tropospheric nitrous acid, Nature Geosci., 4, 440-443, 10.1038/NGEO1164, 2011.

Zhu, C., Xiang, B.. Chu, L.T., and Zhu, L.: 308 nm Photolysis of nitric acid in the gas phase, on aluminum surfaces, and on ice films, J. Phys. Chem. A, 114, 2561-2568, 2010.

Zhu, L., Sangwan, M., Huang, L., Du, J., and Chu, L.T.: Photolysis of nitric acid at 308 nm in the absence and in the presence of water vapor, J. Phys. Chem. A 2015, 119, 4907-4914, 2015.

---

## Author Comment (AC3) · 3 May 2018

Response to the Interactive comment by Anonymous Referee #1 on "Tropospheric HONO Distribution and Chemistry in the Southeast U.S."

C. Ye (c.ye@pku.edu.cn) and X. Zhou (xianliang.zhou@health.ny.gov)

We thank the Anonymous Referee #1 for pointing out the shortcomings in our analysis and presentation of the data. We have significantly revised the manuscript accordingly to address the referee's comments and concerns. Here are our responses to the referee's specific comments.

The referee is correct that $\sim$36% of the original HONO remains after one photolysis lifetime. However, we did not "disregard" the contribution from ground HONO source

simply because HONO photo-lifetime is 8 minutes. We would first like to point out that there has not been any report in literature for 10 ppbv daytime HONO on the ground level, as the referee assumed. The daytime HONO/NOx ratio is in the range of 0.05-0.1 in the low-NOx rural atmosphere (Zhou et al., 2002) and $\sim 0.02$ in high-NOx urban environment (Villena et al., 2011). The direct emission HONO/NOx ratios are even lower, $\leq 0.01$ in automobile exhausts (Kirchstetter et al., 1996; Li et al., 2008) and $\sim 0.002$ in power plant plumes (Neuman et al., 2016). The NOx levels that we observed in the Southeast U.S. was mostly under 0.5 ppbv in the background area and even in the urban plumes; the initial HONO concentration associated with the observed levels of NOx would be $\leq 50$ pptv even if an upper limit HONO/NOx ratio of 0.1 is assumed. With a transport time of $\sim 1$ h from the source, i.e., $\sim 5$ times of the HONO photolysis lifetime, the contribution from the source would be well below the detection limit of 1 pptv of our HONO instrument. Therefore, we argued that the measured HONO was mostly produced in situ from precursors such as NOx and pNO3 within the air masses during the transport, not from the ground HONO sources.

Our conclusion of no significant contribution from the ground HONO source was also based on the vertical profile of HONO. We examined the vertical profiles of HONO, its precursor, isoprene and potential temperature (Figure 4) in section "3.2 HONO contribution from ground-level sources". Isoprene is a biogenic VOC emitted from ground vegetation (trees) and has a lifetime of $\sim 1$ hr. Based on isoprene vertical distribution information, a TBL mixing time can be estimated. The photolysis lifetime of HONO was much shorter than that of isoprene during the daytime NOMADSS flights. If the ground source contributed significant to the TBL HONO budget, a much steeper vertical concentration gradient than that of isoprene should be expected. However, we observed relatively uniform vertical concentration profiles (Figs. 3 and 4), suggesting that contribution from ground HONO source was not important.

We did not randomly assign an air mass as "background" or "urban" just based on HONO concentrations. In the four daytime research flights (RFs #4, #5, #11 and #17)

reported in this manuscript, we conducted our airborne HONO measurements mostly over the rural regions in Southeast U.S., and only sampled urban and power plumes sometimes in RF #11 (Figure 2). A large suite of chemical and metrological parameters were measured onboard the C-130. The identifications of plumes and background air masses were done with the help of plumes tracers like NOx, and benzene (Figure 7) in original manuscript, further with SO2, CO, and acetonitrile (Figure S1) in the revised manuscript. And in RF #18, we performed back trajectory calculations to examine the impact of urban plume from Nashville metropolitan area. Indeed, HONO was being produced from its precursors (including the OH-NO reaction the referee mentioned) in the air mass during the transport. The HONO is considered in situ produced, not directly emitted. When an air mass was influenced by the urban emission, concentrations of HONO precursors and urban tracers (NOx, CO, benzene) were higher, and in situ HONO production would be higher.

We have significantly revised the discussion in the manuscript in both "3.2 HONO contribution from ground-level sources" and "3.4 HONO chemistry in plumes", to address referee's concerns and to make our argument more clearly.

We disagree with the referee that our conclusion was based on weak correlation. The conclusion that inorganic particulate nitrate (pNO3) photolysis is a major HONO source in the air column in Southeast U.S. was first based on directly field observation and HONO budget analysis. With comprehensive parameters related to HONO chemistry were directly measured in our study, we were able to conduct HONO budget analysis. The analysis suggested that known NOx-related reactions can only sustain a minor fraction of the observed HONO source and there was a major fraction of HONO source strength unaccounted in the air column. If particulate nitrate behaves similarly to surface HNO3 photochemically, i.e., with a photolysis rate constant 2-3 order of magnitude higher than that in the gas phase (e.g., Baergen and Donaldson, 2013; Reed et al., 2017; Ye et al., 2016b, 2017a; Zhou et al., 2003, 2011; Zhu et al, 2010, 2015), it could be a potentially important HONO precursor. To examine the potential role of

pNO3 as a HONO precursor, we collected aerosol samples on Teflon filters on the C-130 during the NOMADSS field study and determined the photolysis rate constants of particulate nitrate leading to the productions of HONO (a major product) and NO2 (a minor product). High and highly variable JNpNO3 values were obtained, from $8.3 \times 10{-5}$ s-1 to $3.1 \times 10{-4}$ s-1, with a median of $2.0 \times 10{-4}$ s-1 and a mean ($\pm 1$ standard deviation) of $1.9$ ($\pm 1.2$) $\times 10{-4}$ s-1, when normalized to tropical noontime conditions at ground level (solar zenith angle = 0 o) (Ye et al., 2017b). The laboratory measurement of JpNO3 has been described and discussed in detail in our previous paper (Ye et al., 2017b). HONO budget analysis using the median $J\_{(pNO\_3)}\hat{N}$ value of $2.0 \times 10{-4}$ s-1 suggests that pNO3 photolysis can account for most of the remaining HONO source strength (the original Figures 6b and 7b, and now the revised Figures 5b and 6). The correlation between HONO and its potential precursor pNO3 (the original Figure 5, and now the revised Figure S1) is quite weak (r2 $\sim$0.17), as pointed out by the referee. The correlation between the required HONO source and the contribution from particulate nitrate photolysis ([pNO3]$\times$JpNO3) improved somewhat (r2 = 0.34) (the original Figures 6b, now the revised Figure 5b), but is still not as strong as to be expected from pNO3 photolysis being the major HONO source. As we explained in the manuscript (lines 338-343), "It may be in part due to the use of a single median $J\_pNO3\hat{N}$value of $\sim 2.0 \times 10{-4}$ s-1 in the calculations of the ambient $J\_(ãĂŰpNOãĂŮ\_3$ ) and the production rates of HONO in Figure 5b; the actual $J\_pNO3\hat{N}$values are highly variable, ranging from $8.3 \times 10{-5}$ s-1 to $3.1 \times 10{-4}$ s-1 (Ye et al., 2017). HONO source contribution from particulate nitrate photolysis in Figure 5b are thus estimates of the in situ HONO production rates from pNO3 photolysis in different air masses."

The photolysis lifetime of pNO3 was short using the median value of laboratory determined JpNO3, as the referee pointed out. Many laboratory and field studies have shown the high photolysis rate constant of surface HNO3 (Baergen and Donaldson, 2013; Ye et al., 2016a; Zhou et al., 2003, 2011; Zhu et al, 2010, 2015) and pNO3 (Reed et al.; Ye et al., 2017a, 2017b), lending support to our argument that pNO3 photolysis can be an effective renoxification pathway to recycle nitric acid to photochemically reactive NOx and HONO. However, we would like to point out that particulate nitrate is in a dynamic equilibrium with gas-phase HNO3, and that the later accounts for a larger (or even dominant) fraction of total nitrate (pNO3+HNO3) and is photochemically inert. The overall photolysis of total nitrate (pNO3+HNO3) would be much slower than indicated by JpNO3. In addition, oxidation of NOx via several pathways will replenish the pNO3+HNO3 reservoir. The results reported in this manuscript and in earlier papers (Reed et al., 2017; Ye et al., 2016a) suggest that there is an effective cycling in reactive nitrogen species in the low-NOx atmosphere, sustaining the observed levels of HONO and pNO3.

Some mechanisms have been proposed to explain the large enhancement of photolysis rate constant for surface HNO3 and pNO3, by 2-3 orders of magnitude compared to that of gas-phase HNO3. The light absorption by HNO3 in the UV range has been found to be 1-4 orders of magnitude higher on surfaces of silicon and ice than in the gas phase, with a significant red shift to long wavelength (Du et al., 2011; Zhu et al., 2008, 2015), probably resulting from bond stretching and/or bond deformation (Svoboda et al., 2013). Since the photolysis yield stays relatively high, 0.8-0.9 (Zhu et al, 2010), the resulting effect of the catalytic surface is the enhancement of photolysis rate constant over that in gas phase. In addition, organic and inorganic chromophores on ambient surfaces and in aerosol particles can enhance the photolysis of the associated HNO3 and nitrate through photosensitization (Ye et al., 2016b, 2017b). We also hypothesized that NO2 is the dominant primary product of the photolysis of surface HNO3 and pNO3, and the produced NO2 (adsorbed) may react quickly with organics and water molecules on the surface and in aerosol particles to produce HONO as the secondary product. The proposed mechanism explains the laboratory results showing NO2 as the dominant product from HNO3 photolysis on clean and dry laboratory surfaces (Zhou et al., 2003; Zhu et al., 2010), while HONO as the major product on natural surfaces and in ambient aerosols (Ye et al., 2016a, 2017b).

We have not got the chance to compare our LPAS pNO3 method with other instruments. We would like to do that as soon as we have the opportunity.

As to the issue of power plant plumes raised by the referee, the answer is similar to what we just provided for urban plumes. That is, the power plant plumes we encountered were small and relatively diluted with NOx levels up to 1.8 ppbv (Figures 2 and 7), and the directly emitted HONO would be ≤ 18 pptv ppbv even if a high HONO/NOx emission ratio of 0.01 is assumed (Neuman et al., 2016). During the 1-h transport time, the remaining HONO concentration from the direct emission would be well below our detection limit of 1 pptv. Therefore, the HONO concentrations in the power plant plumes were mostly produced within the air mass during the transportation, from elevated HONO precursors from anthropogenic sources. In fact, the HONO measured in fresh power plant plumes has been found to be mostly secondarily produced (Neuman et al., 2016).

Lastly, we regret the omission to reference the recent paper by Neuman et al. (2016). We finished our first draft of this manuscript over two years ago, before the publication of the mentioned paper. Although we have made significant changes to the draft during the subsequent revisions, we failed to update the references. The paper by Neuman et al. (2016) has been referenced and discussed in the revised manuscript.

References

Baergen, A. M., and Donaldson, D. J.: Photochemical renoxification of nitric acid on real urban grime, Environ. Sci. Technol., 47, 815-820, 10.1021/es3037862, 2013.

Du, J., and Zhu, L.: Quantification of the absorption cross sections of surface-adsorbed nitric acid in the 335-365 nm region by Brewster angle cavity ring-down spectroscopy, Chem. Phys. Lett., 511, 213-218, 10.1016/j.cplett.2011.06.062, 2011.

Kirchstetter, T. W, Harley,R.A., and Littejohn, D.: Measurements of nitrous acid in motor vehicle exhaust, Environ. Sci. Technol., 30 (9), 2843–2849, 1996.

Li, Y. Q., Schwab, J. J., and Demerjian, K. L.: Fast time response measurements

of gaseous nitrous acid using a tunable diode laser absorption spectrometer: HONO emission source from vehicle exhausts, Geophys. Res. Lett., 35, 2008.

Neuman, J. A., et al.: HONO emission and production determined from airborne measurements over the Southeast U.S., J. Geophys. Res. Atmos., 121, 9237–9250, doi:10.1002/2016JD025197, 2016 Reed, C. et al.: Evidence for renoxification in the tropical marine boundary layer, Atmos. Chem. Phys., 17, 4081–4092, 2017.

Svoboda, O.; Kubelova, L.; Slavicek, P.: Enabling forbidden processes: Quantum and solvation enhancement of nitrate anion UV absorption, J. Phys. Chem. A, 117, 12868-12877, 2013.

Villena, G., Kleffmann, J., Kurtenbach, R., Wiesen, P., Lissi, E., Rubio, M. A., Croxatto, G., and Rappengluck, B.: Vertical gradients of HONO, NOx and O3 in Santiago de Chile, Atmos. Environ., 45, 3867-3873, 2011.

Ye, C., et al.: Rapid cycling of reactive nitrogen in the marine boundary layer, Nature, 532, 489-491, 2016a.

Ye, C., Gao, H., Zhang, N., and Zhou, X.: Photolysis of nitric Acid and nitrate on natural and artificial surfaces, Environ. Sci. Technol., 50, 3530-3536, 2016b.

Ye, C., Heard, D.E., and Whalley, L.K.: Evaluation of novel routes for NOx formation in remote regions, Environ. Sci. Technol., DOI: 10.1021/acs.est.6b06441, 2017a.

Ye, C., Zhang, N., Gao, H., and Zhou, X.: Photolysis of particulate nitrate as a source of HONO and NOx, Environ. Sci. Technol., DOI: 10.1021/acs.est.7b00387, 2017b.

Zhou, X., Civerolo, K., Dai, H., Huang, G., Schwab, J., and Demerjian, K.: Summertime nitrous acid chemistry in the atmospheric boundary layer at a rural site in New York State, J. Geophys. Res., 107, doi:10.1029/2001JD001539, 2002.

Zhou, X., Gao, H., He, Y., Huang, G., Bertman, S. B., Civerolo, K., and Schwab, J.: Nitric acid photolysis on surfaces in low-NOx environments: Significant atmospheric

implications, Geophys. Res. Lett., 30, 2217, 10.1029/2003gl018620, 2003.

Zhou, X., Zhang, N., TerAvest, M., Tang, D., Hou, J., Bertman, S., Alaghmand, M., Shepson, P. B., Carroll, M. A., Griffith, S., Dusanter, S., and Stevens, P. S.: Nitric acid photolysis on forest canopy surface as a source for tropospheric nitrous acid, Nature Geosci., 4, 440-443, 10.1038/NGEO1164, 2011.

Zhu, C. Z., Xiang, B., Zhu, L., and Cole, R.: Determination of absorption cross sections of surface-adsorbed HNO3 in the 290-330 nm region by Brewster angle cavity ring-down spectroscopy, Chem. Phys. Lett., 458, 373-377, 2008.

Zhu, C., Xiang, B., Chu, L.T., and Zhu, L.: 308 nm Photolysis of nitric acid in the gas phase, on aluminum surfaces, and on ice films, J. Phys. Chem. A, 114, 2561-2568, 2010.

Zhu, L., Sangwan, M., Huang, L., Du, J., and Chu, L.T.: Photolysis of nitric acid at 308 nm in the absence and in the presence of water vapor, J. Phys. Chem. A, 119, 4907-4914, 2015.

Please also note the supplement to this comment:
https://www.atmos-chem-phys-discuss.net/acp-2018-105/acp-2018-105-AC3-supplement.pdf

––––––––––––––––––––––––––––––––

**Supplement:**

**Response to the Interactive comment by Anonymous Referee #1 on "Tropospheric HONO Distribution and Chemistry in the Southeast U.S."**

C. Ye (c.ye@pku.edu.cn) and X. Zhou (xianliang.zhou@health.ny.gov)

*The manuscript "Tropospheric HONO distribution and chemistry in the Southeast U.S." by Ye et al. presents HONO measurements made during the NOMADSS campaign. The two main claims presented here are: (1) there is more HONO observed than can be explained by known chemistry, and (2) photolysis of particle nitrate accounts for this so-called missing HONO source. The analysis used to make both claims are weak, therefore, unconvincing. Moreover, the analysis on nighttime chemistry and production in power plant exhaust were hastily done and written. This work can be considered for publication only after significant improvements.*

**Response:** We thank the Anonymous Referee #1 for pointing out the shortcomings in our analysis and presentation of the data. We have significantly revised the manuscript accordingly to address the referee's comments and concerns. Here are our responses to the referee's specific comments.

*As for (1), the authors claim that because HONO photo-lifetime is 8 minutes, that direct emission of HONO can be disregarded. This is a misinterpretation of the concept of lifetime. Lifetime represents an e-folding time, meaning that ~36% of the original amount still remains after 8 minutes since time zero, or time since emission. If HONO at the emission source is 10 ppb, approximately 50 to 60 minutes is required for HONO to reach 11 ppt, the median HONO level reported. Judging by figure 2, the median HONO value of 11 ppt (what the authors claim is anthropogenic-free HONO) is observed in close proximity to urban plumes (20 to 30 ppt, identified in figure 2). This 50 to 60 minute period is an underestimate since it does not account for re-formation of HONO by OH + NO, both of which are likely to be elevated in urban plumes. Bottom line is that the authors need to demonstrate convincingly that the 11 ppt HONO is not derived from anthropogenic sources (by quantitatively accounting for mixing, emissions, and chemistry), because the case for this so-called 'extra' HONO is the difference between 11 ppt and 2 ppt (amount of HONO expected assuming PSS without this 'extra' source). The analysis as it currently stands in inadequate.*

**Response:** The referee is correct that ~36% of the original HONO remains after one photolysis lifetime. However, we did not "disregard" the contribution from ground HONO source simply because HONO photo-lifetime is 8 minutes. We would first like to point out that there has not been any report in literature for 10 ppbv daytime HONO on the ground level, as the referee assumed. The daytime HONO/$NO_x$ ratio is in the range of 0.05-0.1 in the low-$NO_x$ rural atmosphere (Zhou et al., 2002) and ~ 0.02 in high-$NO_x$ urban environment (Villena et al., 2011). The direct emission HONO/$NO_x$ ratios are even lower, ≤0.01 in automobile exhausts (Kirchstetter et al., 1996; Li et al., 2008) and ~0.002 in power plant plumes (Neuman et al., 2016). The $NO_x$ levels that we observed in the Southeast U.S. was mostly under 0.5 ppbv in the background area and even in the urban plumes; the initial HONO concentration associated with the observed levels of $NO_x$ would be ≤ 50 pptv even if an upper limit HONO/$NO_x$ ratio of 0.1 is assumed. With a transport time of ~1 h from the source, i.e., ~ 5 times of the HONO photolysis lifetime, the contribution from the source would be well below the detection limit of 1 pptv of our HONO instrument. Therefore, we argued that the measured HONO was mostly produced *in*

*situ* from precursors such as $NO_x$ and $pNO_3$ within the air masses during the transport, not from the ground HONO sources.

Our conclusion of no significant contribution from the ground HONO source was also based on the vertical profile of HONO. We examined the vertical profiles of HONO, its precursor, isoprene and potential temperature (Figure 4) in section "3.2 HONO contribution from ground-level sources". Isoprene is a biogenic VOC emitted from ground vegetations (trees) and has a lifetime of ~1 hr. Based on isoprene vertical distribution information, a TBL mixing time can be estimated. The photolysis lifetime of HONO was much shorter than that of isoprene during the daytime NOMADSS flights. If the ground source contributed significant to the TBL HONO budget, a much steeper vertical concentration gradient than that of isoprene should be expected. However, we observed relatively uniform vertical concentration profiles (Figs. 3 and 4), suggesting that contribution from ground HONO source was not important.

We did not randomly assign an air mass as "background" or "urban" just based on HONO concentrations. In the four daytime research flights (RFs #4, #5, #11 and #17) reported in this manuscript, we conducted our airborne HONO measurements mostly over the rural regions in Southeast U.S., and only sampled urban and power plumes sometimes in RF #11 (Figure 2). A large suite of chemical and metrological parameters were measured onboard the C-130. The identifications of plumes and background air masses were done with the help of plumes tracers like $NO_x$, and benzene (Figure 7) in original manuscript, further with $SO_2$, CO, and acetonitrile (Figure S1) in the revised manuscript. And in RF #18, we performed back trajectory calculations to examine the impact of urban plume from Nashville metropolitan area. Indeed, HONO was being produced from its precursors (including the OH-NO reaction the referee mentioned) in the air mass during the transport. The HONO is considered *in situ* produced, not directly emitted. When an air mass was influenced by the urban emission, concentrations of HONO precursors and urban tracers ($NO_x$, CO, benzene) were higher, and *in situ* HONO production would be higher.

We have significantly revised the discussion in the manuscript in both "3.2 HONO contribution from ground-level sources" and "3.4 HONO chemistry in plumes", to address referee's concerns and to make our argument more clearly.

*As for (2), the authors conclude a causal relationship between photolysis of particulate nitrate and HONO based on rather weak correlation (figure 5). That is less than convincing. Moreover, a photolysis rate of 2e-4 sec-1 means a photo-lifetime less than 1.5 hours for particulate nitrates. What are these nitrates? inorganic or organic? Has there been any reports of particle-phase nitrates exhibiting photo-lifetimes on the order of 1.5 hours? Is there a mechanism proposed? What remains in the particle-phase as the nitrate is released as HONO? Is all of the nitrate turn into HONO, or NO or NO2 or HNO3? I am concerned the photolysis conducted in the laboratory is not atmospherically relevant. More information on this lab photolysis experiment may help. How do the nitrate abundances measured with this filter method compare to what other instruments (AMS? PILS?) have measured for particle nitrates? And as for the power plant analysis, the same concerns I have for claim (1) applies here. You cannot assume just because the plume has been transported over 1 hour that none of the HONO observed is anthropogenic in origin. You need to know what the mixing ratio was near the emission point to know whether the HONO measured downwind was or was not directly emitted because the photo-lifetime is an e-folding time, it does not just disappear after 8 minutes. Lastly, citing previous work on the*

*subject could be useful. Recent work by Neuman et al. 2016 comes to mind (https://agupubs.onlinelibrary.wiley.com/doi/abs/10.1002/2016JD025197).*

**Response:** We disagree with the referee that our conclusion was based on weak correlation. The conclusion that inorganic particulate nitrate (pNO$_3$) photolysis is a major HONO source in the air column in Southeast U.S. was first based on directly field observation and HONO budget analysis. With comprehensive parameters related to HONO chemistry were directly measured in our study, we were able to conduct HONO budget analysis. The analysis suggested that known NO$_x$-related reactions can only sustain a minor fraction of the observed HONO source and there was a major fraction of HONO source strength unaccounted in the air column. If particulate nitrate behaves similarly to surface HNO$_3$ photochemically, i.e., with a photolysis rate constant 2-3 order of magnitude higher than that in the gas phase (e.g., Baergen and Donaldson, 2013; Reed et al., 2017; Ye et al., 2016b, 2017a; Zhou et al., 2003, 2011; Zhu et al, 2010, 2015), it could be a potentially important HONO precursor. To examine the potential role of pNO$_3$ as a HONO precursor, we collected aerosol samples on Teflon filters on the C-130 during the NOMADSS field study and determined the photolysis rate constants of particulate nitrate leading to the productions of HONO (a major product) and NO$_2$ (a minor product). High and highly variable $J^N_{pNO3}$ values were obtained, from $8.3 \times 10^{-5}$ s$^{-1}$ to $3.1 \times 10^{-4}$ s$^{-1}$, with a median of $2.0 \times 10^{-4}$ s$^{-1}$ and a mean ($\pm$ 1 standard deviation) of $1.9 (\pm 1.2) \times 10^{-4}$ s$^{-1}$, when normalized to tropical noontime conditions at ground level (solar zenith angle = 0$^{\circ}$) (Ye et al., 2017b). The laboratory measurement of $J_{pNO3}$ has been described and discussed in detail in our previous paper (Ye et al., 2017b). HONO budget analysis using the median $J^N_{pNO_3}$ value of $2.0 \times 10^{-4}$ s$^{-1}$ suggests that pNO$_3$ photolysis can account for most of the remaining HONO source strength (the original Figures 6b and 7b, and now the revised Figures 5b and 6).

The correlation between HONO and its potential precursor pNO$_3$ (the original Figure 5, and now the revised Figure S1) is quite weak (r$^2$ ~0.17), as pointed out by the referee. The correlation between the required HONO source and the contribution from particulate nitrate photolysis ([pNO$_3$]$\times J_{pNO3}$) improved somewhat (r$^2$ = 0.34) (the original Figures 6b, now the revised Figure 5b), but is still not as strong as to be expected from pNO$_3$ photolysis being the major HONO source. As we explained in the manuscript (lines 338-343), "It may be in part due to the use of a single median $J^N_{pNO3}$ value of ~ $2.0 \times 10^{-4}$ s$^{-1}$ in the calculations of the ambient $J_{pNO_3}$ and the production rates of HONO in Figure 5b; the actual $J^N_{pNO3}$ values are highly variable, ranging from $8.3 \times 10^{-5}$ s$^{-1}$ to $3.1 \times 10^{-4}$ s$^{-1}$ (Ye et al., 2017). HONO source contribution from particulate nitrate photolysis in Figure 5b are thus estimates of the *in situ* HONO production rates from pNO$_3$ photolysis in different air masses."

The photolysis lifetime of pNO$_3$ was short using the median value of laboratory determined $J_{pNO3}$, as the referee pointed out. Many laboratory and field studies have shown the high photolysis rate constant of surface HNO$_3$ (Baergen and Donaldson, 2013; Ye et al., 2016a; Zhou et al., 2003, 2011; Zhu et al, 2010, 2015) and pNO$_3$ (Reed et al.; Ye et al., 2017a, 2017b), lending support to our argument that pNO$_3$ photolysis can be an effective renoxification pathway to recycle nitric acid to photochemically reactive NO$_x$ and HONO. However, we would like to point out that particulate nitrate is in a dynamic equilibrium with gas-phase HNO$_3$, and that the later accounts for a larger (or even dominant) fraction of total nitrate (pNO$_3$+HNO$_3$) and is photochemically inert. The overall photolysis of total nitrate (pNO$_3$+HNO$_3$) would be much slower than indicated by $J_{pNO3}$. In addition, oxidation of NO$_x$ via several pathways will replenish the pNO$_3$+HNO$_3$ reservoir. The results reported in this manuscript and in earlier papers (Reed et

al., 2017; Ye et al., 2016a) suggest that there is an effective cycling in reactive nitrogen species in the low-$NO_x$ atmosphere, sustaining the observed levels of HONO and $pNO_3$.

Some mechanisms have been proposed to explain the large enhancement of photolysis rate constant for surface $HNO_3$ and $pNO_3$, by 2-3 orders of magnitude compared to that of gas-phase $HNO_3$. The light absorption by $HNO_3$ in the UV range has been found to be 1-4 orders of magnitude higher on surfaces of silicon and ice than in the gas phase, with a significant red shift to long wavelength (Du et al., 2011; Zhu et al., 2008, 2015), probably resulting from bond stretching and/or bond deformation (Svoboda et al., 2013). Since the photolysis yield stays relatively high, 0.8-0.9 (Zhu et al, 2010), the resulting effect of the catalytic surface is the enhancement of photolysis rate constant over that in gas phase. In addition, organic and inorganic chromophores on ambient surfaces and in aerosol particles can enhance the photolysis of the associated $HNO_3$ and nitrate through photosensitization (Ye et al., 2016b, 2017b). We also hypothesized that $NO_2$ is the dominant primary product of the photolysis of surface $HNO_3$ and $pNO_3$, and the produced $NO_2$ (adsorbed) may react quickly with organics and water molecules on the surface and in aerosol particles to produce HONO as the secondary product. The proposed mechanism explains the laboratory results showing $NO_2$ as the dominant product from $HNO_3$ photolysis on clean and dry laboratory surfaces (Zhou et al., 2003; Zhu et al., 2010), while HONO as the major product on natural surfaces and in ambient aerosols (Ye et al., 2016a, 2017b).

We have not got the chance to compare our LPAS $pNO_3$ method with other instruments. We would like to do that as soon as we have the opportunity.

As to the issue of power plant plumes raised by the referee, the answer is similar to what we just provided for urban plumes. That is, the power plant plumes we encountered were small and relatively diluted with $NO_x$ levels up to 1.8 ppbv (Figures 2 and 7), and the directly emitted HONO would be ≤ 18 pptv ppbv even if a high HONO/$NO_x$ emission ratio of 0.01 is assumed (Neuman et al., 2016). During the 1-h transport time, the remaining HONO concentration from the direct emission would be well below our detection limit of 1 pptv. Therefore, the HONO concentrations in the power plant plumes were mostly produced within the air mass during the transportation, from elevated HONO precursors from anthropogenic sources. In fact, the HONO measured in fresh power plant plumes has been found to be mostly secondarily produced (Neuman et al., 2016).

Lastly, we regret the omission to reference the recent paper by Neuman et al. (2016). We finished our first draft of this manuscript over two years ago, before the publication of the mentioned paper. Although we have made significant changes to the draft during the subsequent revisions, we failed to update the references. The paper by Neuman et al. (2016) has been referenced and discussed in the revised manuscript.

**References**
Baergen, A. M., and Donaldson, D. J.: Photochemical renoxification of nitric acid on real urban grime, Environ. Sci. Technol., 47, 815-820, 10.1021/es3037862, 2013.
Du, J., and Zhu, L.: Quantification of the absorption cross sections of surface-adsorbed nitric acid in the 335-365 nm region by Brewster angle cavity ring-down spectroscopy, Chem. Phys. Lett., 511, 213-218, 10.1016/j.cplett.2011.06.062, 2011.
Kirchstetter, T. W, Harley,R.A., and Littejohn, D.: Measurements of nitrous acid in motor vehicle exhaust, Environ. Sci. Technol., 30 (9), 2843–2849, 1996.

Li, Y. Q., Schwab, J. J., and Demerjian, K. L.: Fast time response measurements of gaseous nitrous acid using a tunable diode laser absorption spectrometer: HONO emission source from vehicle exhausts, Geophys. Res. Lett., 35, 2008.

Neuman, J. A., et al.: HONO emission and production determined from airborne measurements over the Southeast U.S., J. Geophys. Res. Atmos., 121, 9237–9250, doi:10.1002/2016JD025197, 2016

Reed, C. et al.: Evidence for renoxification in the tropical marine boundary layer, *Atmos. Chem. Phys.*, 17, 4081–4092, 2017.

Svoboda, O.; Kubelova, L.; Slavicek, P.: Enabling forbidden processes: Quantum and solvation enhancement of nitrate anion UV absorption, J. Phys. Chem. A, 117, 12868-12877, 2013.

Villena, G., Kleffmann, J., Kurtenbach, R., Wiesen, P., Lissi, E., Rubio, M. A., Croxatto, G., and Rappengluck, B.: Vertical gradients of HONO, $NO_x$ and $O_3$ in Santiago de Chile, Atmos. Environ., 45, 3867-3873, 2011.

Ye, C., et al.: Rapid cycling of reactive nitrogen in the marine boundary layer, Nature, 532, 489-491, 2016a.

Ye, C., Gao, H., Zhang, N., and Zhou, X.: Photolysis of nitric Acid and nitrate on natural and artificial surfaces, Environ. Sci. Technol., 50, 3530-3536, 2016b.

Ye, C., Heard, D.E., and Whalley, L.K.: Evaluation of novel routes for NOx formation in remote regions, Environ. Sci. Technol., DOI: 10.1021/acs.est.6b06441, 2017a.

Ye, C., Zhang, N., Gao, H., and Zhou, X.: Photolysis of particulate nitrate as a source of HONO and $NO_x$, Environ. Sci. Technol., DOI: 10.1021/acs.est.7b00387, 2017b.

Zhou, X., Civerolo, K., Dai, H., Huang, G., Schwab, J., and Demerjian, K.: Summertime nitrous acid chemistry in the atmospheric boundary layer at a rural site in New York State, J. Geophys. Res., 107, doi:10.1029/2001JD001539, 2002.

Zhou, X., Gao, H., He, Y., Huang, G., Bertman, S. B., Civerolo, K., and Schwab, J.: Nitric acid photolysis on surfaces in low-NOx environments: Significant atmospheric implications, Geophys. Res. Lett., 30, 2217, 10.1029/2003gl018620, 2003.

Zhou, X., Zhang, N., TerAvest, M., Tang, D., Hou, J., Bertman, S., Alaghmand, M., Shepson, P. B., Carroll, M. A., Griffith, S., Dusanter, S., and Stevens, P. S.: Nitric acid photolysis on forest canopy surface as a source for tropospheric nitrous acid, Nature Geosci., 4, 440-443, 10.1038/NGEO1164, 2011.

Zhu, C. Z., Xiang, B., Zhu, L., and Cole, R.: Determination of absorption cross sections of surface-adsorbed $HNO_3$ in the 290-330 nm region by Brewster angle cavity ring-down spectroscopy, Chem. Phys. Lett., 458, 373-377, 2008.

Zhu, C., Xiang, B., Chu, L.T., and Zhu, L.: 308 nm Photolysis of nitric acid in the gas phase, on aluminum surfaces, and on ice films, J. Phys. Chem. A, 114, 2561-2568, 2010.

Zhu, L., Sangwan, M., Huang, L., Du, J., and Chu, L.T.: Photolysis of nitric acid at 308 nm in the absence and in the presence of water vapor, J. Phys. Chem. A, 119, 4907-4914, 2015.

---

## Author Comment (AC4) · 3 May 2018

Response to the Interactive comment by Anonymous Referee #2 on "Tropospheric HONO Distribution and Chemistry in the Southeast U.S."

C. Ye (c.ye@pku.edu.cn) and X. Zhou (xianliang.zhou@health.ny.gov)

"General Comments This manuscript explores the generation and fate of HONO above and within the planetary boundary layer over the southeastern United States during NOMADSS 2013 from several research flights aboard the NCAR C-130 aircraft. The vertical distribution of HONO throughout this layer is clearly demonstrated to be derived from volume sources, with a robust testing of the known mechanisms of HONO formation against parameterizations of particulate nitrate photolysis, which is emerging as an important source of tropospheric HONO. The Author's find that previously established volume-based mechanisms of HONO formation cannot account for the observed quantities and that the photolysis nitrate in the condensed phase can possibly explain the majority of the observed quantities. The Authors demonstrate that HONO is a minor OH source at these altitudes when its production is driven solely from volume production and also that it is an important intermediate in the renoxification pathways of tropospheric trans- port of nitrogen oxides. Overall, this manuscript is well written with a solid analysis of the dataset. There are minor modifications necessary to make the manuscript more clear and concise in its purpose and findings. The removal of some figures and text by the production of a supporting information document would easily facilitate this."

Response: We thank Anonymous Referee #2 for the positive and encouraging comments. The detailed and insightful comments and suggestions have greatly helped us in revising and improving the manuscript. A supplement document has been generated to include supporting content as suggested.

"Specific Comments Page 2, Lines 7-10: The detailed analysis of the isoprene transport and subsequent lifetime calculations for HONO are a quantitative assessment of the decoupling of surface emissions from the observed HONO. The Authors should consider using their quantitative assessment as the basis for their statement here instead of the more qualitative observation of no vertical gradient."

Response: The abstract has been revised and reorganized, as the referee suggested.

"Page 2, Lines 14-15: Please provide the average +/- SD of the actual fraction of the observed HONO that was generated by pNO3 photolysis from the presented calculations instead of 'appeared to be the major daytime HONO source'."

Response: The abstract has been revised and reorganized, as the referee suggested.

"Page 2, Lines 20-25: Provide the quantitative findings from each section of the detailed analysis here over the general statements of relative importance. This will generate greater impact for this work."

Response: The abstract has been revised and reorganized, as the referee suggested.

"Page 3, Line 39: Remove ', as an important OH precursor,' as it is redundant."

Response: The phrase has been removed as suggested.

"Page 3, Lines 51-57: I would suggest removing this length section and replacing it with a single sentence following the statements on combustion HONO sources (Line 48). This level of detail in the introduction is not really relevant to the tropospheric chemistry discussed in this work."

Response: Revision has made as the referee suggested.

"Page 4, Lines 75-84: The last two sentences demonstrate that R4 is unnecessary and it should likely be removed from here and from the presented data analysis, since it has been show to be a two-photon process. It should be removed here and the section on the hydroperoxyl-water complex mechanisms should be replaced with one sentence on its existence and low yield of HONO."

Response: As the referee pointed out, both reactions (R4) and (R5) are not important as HONO sources. Since we intend to conduct HONO budget analyses in the later sections, to include all NOx-related reactions. We feel that a brief discussion of reactions (R4) and (R5) here is justified.

"Page 5, Lines 104-107: It would be useful to guide the readers through the major explorations of this dataset here. Consider listing the major sections of this work here in the order that they are presented in the abstract, manuscript, and conclusions, to improve clarity."

Response: We did provide some field campaign and measurement information in the first paragraph in section "2 Experimental", right after the paragraph.

"Page 5, Line 108: The experimental section could use subsections to improve clarity."

Response: Revision has made as the referee suggested.

"Page 5, Line 126: The baseline subtraction of interferences from particulate nitrite here does not acknowledge that there is a size-dependent collection efficiency in these style of instruments. For example, fog droplets would be effectively captured in the primary channel to appear as HONO and not be corrected for in the secondary channel. This has been demonstrated in other works with this analytical approach (e.g. (Sörgel et al., 2011) and references therein). Is there any potential for droplet nitrite interferences in these measurements where clouds may have been encountered?"

Response: As pointed out by the referee, cloud droplets may be collected at significant efficiency and be an interference in our measurement. We have excluded the in-cloud measurement data from our data analysis, due to lack of valid way to correct the data. The following sentences have been added in the revised manuscript (lines 163-166): "Noisy baselines were observed when the C-130 was flying in the clouds, due to the sampling of cloud droplets by our sampling systems. Because of the lack of a valid way to correct for this interference, all in-cloud measurement data of HONO and pNO3 have been excluded from the data analysis."

"Page 6, Lines 138-139: It is confusing to follow the logic of this estimation. Was the maximum possible interference determined in some sections of the dataset to set the limit at 0.2? If possible, add the quantitative approach used to a section in a Supporting Information document. If not, please improve the clarity here."

Response: We have not determined the collection efficiency for $HO_2NO_2$ experimentally. The upper limit $HO_2NO_2$-to-HONO conversion efficiency of 0.2 was estimated from the ratio of the observed [HONO] to the calculated $[HO_2NO_2]_{SS}$ in cold, high altitude air masses under our measurement conditions, assuming ambient HONO concentration approaching zero. We found that the correction was not necessary in the TBL. The discussion has been revised to: "Potential interference from peroxynitric acid

(HO2NO2) was suppressed by heating the PFA sampling line to 50 °C. The HO2NO2 steady state concentration ([HO2NO2]SS) was estimated to be less than 1 pptv at temperatures of 20 - 30 °C in the background PBL (Gierczak et al., 2005), and thus interference from HO2NO2 was negligible. Whereas in power plant and urban plumes in the PBL or biomass burning plumes in the upper free troposphere (FT), HO2NO2 interference was not negligible and thus a correction for HONO measurement was made. An upper-limit HO2NO2 response efficiency was estimated to be 0.2 for our HONO measurement systems. The estimation was made from the lowest ratio of the measured HONO to the corresponding [HO2NO2]SS in cold air masses at high altitude, assuming no HONO existed. HONO concentration were then corrected by subtracting a term of "0.2 × [HO2NO2]SS". The correction was below 10% of the measured HONO concentrations in the PBL plumes. However, there may be over-corrections in the cold free troposphere." (lines 13-143)

"Page 6, Lines 142-144: Provide the correlation coefficient, slope, and intercept here to improve clarity and validity of analytical approach."

Response: The intercomparison of HONO measurements from the two instruments (the DOAS and the LPAP) was made by overlaying the concentration time-series on each other (Extended Data Fig. S3 in Ye et al., 2016). The measured concentrations closely tracked each other, and the agreements were within the assessed uncertainties. The readers are encouraged to read the paper for more information.

"Page 6, Line 149: The order of the used apparatus is not clear. Presumably the denuder followed the filter? Please clarify."

Response: As the referee suggested, the sentence has been revised as suggested to ""Zero-pNO3" air was generated to establish measurement baselines for pNO3 by passing the ambient air through a Teflon filter to remove aerosol particles and then a NaCl-coated denuder to remove HNO3 before reaching the sampling unit of LPAP." (lines 155-158).

Interactive
comment

"Page 6, Line 160: Delete 'NCAR's'"

Response: Revision has made as the referee suggested.

"Page 7, Line 161: What are 'state parameter measurements'?"

Response: The NSF/NCAR C-130 aircraft comes equipped with a package of standard instrumentation that flies on all C-130 research missions. The measurements made by these sensors form the core of any research program and provide the information necessary to place the aircraft in space and time while characterizing the basic "state" of the local environment. The parameters include aircraft longitude, latitude, altitude, flight speed, pressure, temperature, dew point, and many more.

"Page 7, Lines 183-184: Remove this from here. It is discussed in sufficient detail later and distracts from the results."

Response: Indeed, the vertical HONO distribution is discussed in the following section in more details. However, we feel that in the "General data description" section, this sentence provides some contrast to the horizontal inhomogeneity of HONO distribution, and thus we keep the sentence as it is.

"Page 7, Lines 186-188: Remove these statements. The information is already presented in the Table and does not need repeating."

Response: The readers can obtain the information directly from these statements without going the tables and Figures. We feel that some degree of redundancy may be needed. Thus we keep the sentence as it is.

"Page 7, Lines 191-192: Delete the sentence on the future paper."

Response: Revision has made as referee #2 suggested.

"Page 8, Lines 194-196: Delete these and direct the reader to the relevant section at the end of the preceding sentence by adding '(Section 3.4)'"

Response: Revision has made as the referee suggested.

"Page 8, Lines 201-203: Remove these statements. The information is already presented in the Table and does not need repeating."

Response: Again, the readers can obtain the information directly from these statements without going the tables and Figures. We feel that some degree of redundancy may be needed. Thus we keep the sentence as it is.

"Page 8, Lines 210-212: Remove these statements. The information is already presented in the Table and does not need repeating."

Response: Again, the readers can obtain the information directly from these statements without going the tables and Figures. We feel that some degree of redundancy may be needed. Thus we keep the sentence as it is.

"Page 9, Line 236: Here is the first definition of the altitudes considered to by the PBL versus the FT. The Authors should add their criteria for distinguishing between the PBL and FT to the methods section. If it would be a lengthy addition, then a condensed description with supporting details could be placed in the Supporting Information document."

Response: The discussion in this section was focused on the transport and contribution of HONO from ground level to the overlying PBL, based on the vertical distributions of HONO and other species in the PBL (300 -1200 m). Transport into the FT would be much slower and was not discussed in the section. The PBL height can be estimated by the temperature inversion in the vertical potential temperature profile.

"Page 9, Lines 238-250: This is a fantastic analysis of the vertical mixing and transport of surface-emitted species, but it is outside the focus of this work. Consider relocating this detailed analysis to the Supporting Information document."

Response: The main discussion of this manuscript is on HONO daytime budget and chemistry. HONO is a unique species mainly produced by heterogeneous processes

on surfaces. Ground surfaces provide the sites for the heterogeneous processes to produce HONO. We feel that it is important to examine the input from ground HONO source to the HONO budget in the PBL. Therefore, we keep the equation (Eq. 1), vertical profiles in Figure 4, and discussion of vertical transport in the main section.

"Page 9, Lines 250-254: Distinguish between ground-emitted and volume-produced HONO here to improve clarity."

Response: We have significantly modified the discussion in that paragraph. The two sentences have been changed to "With a photolytic lifetime of $\sim$ 11 min for HONO, about 11% of the HONO originated from the ground level is expected to reach the altitude of 300 m, the lowest flight altitude of the C-130 aircraft between 11:00 – 12:15 LT in RF #4." (lines 266-268).

"Page 9, Line 256: 'of its precursors' should be 'of its potential precursors' since this work is yet to demonstrate this quantitatively (although it is shown quite well later)."

Response: Revision has made as the referee suggested.

"Page 10, Lines 286-287: This was stated in the introduction as insignificant (and potentially invalid), so why have the authors chosen to include this in their analysis? Suggest removing throughout."

Response: We intended to include in our calculation all the NOx-related reactions reported in literature. While the importance of R4 and R5 are still under debate in literature, our HONO budget analysis does suggest they were insignificant under the conditions we encountered in the Southeast U.S., as stated in the Introduction.

"Pages 10-11, Lines 289-291: Consider providing a justification for selecting all upper limits in these calculations to improve clarity."

Response: As suggested by the referee, the following sentence has been added after equation (Eq. 2): "It should be noted that the upper limit values of rate constants were used in the calculation to avoid the underestimation of [HONO]pss value." (line

307-309)

"Page 11, Line 302: Remove ', such as pNO3.' As it is redundant for the transition between paragraphs."

Response: The phrase has been removed as suggested.

"Page 11, Lines 309-310: Remove 'over the terrestrial areas', 'on Teflon filters. . . summer field study'. This information is already presented in the methods."

Response: The redundant information has been removed as suggested.

"Page 12, Lines 326-330: This is a single sentence and is difficult to follow. Consider breaking into 2-3 sentences to improve clarity."

Response: The long sentence has been changed to "However, the r2 of 0.34 is not as strong as expected from pNO3 photolysis being the major volume HONO source. It may be in part due to the use of a single median J_pNO3ˆN value of $\sim 2.0 \times 10\text{-}4$ s-1 in the calculations of the ambient J_(ãĂŰpNOãĂŮ_3 ) and the production rates of HONO in Figure 5b; the actual J_pNO3ˆN values are highly variable, ranging from 8.3 $\times$ 10-5 s-1 to 3.1 $\times$ 10-4 s-1 (Ye et al., 2017)." (line 337-341)

"Page 12, Line 331: Delete 'only rough'. Redundant. Also see comments on Figure 6 regarding weighted error analysis."

Response: The redundant phrase has been deleted as suggested.

"Page 13, Line 357: What is the error on this ratio of 0.02? Is it statistically different from the fresh power plant emissions?"

Response: Standard deviations of the HONO/NOx ratio have been added in the revised manuscript. The sentence has been revised and expanded to "The observed HONO/NOx ratio was 0.019 $\pm$ 0.004 in the power plant plumes (e.g., P4) and 0.057$\pm$ 0.0019 in urban plumes, significantly higher than the typical HONO/NOx emission ratio of $\sim$0.002 in the fresh power plant plumes (Neuman et al., 2016) and $\leq$0.01 in

automobile exhaust (Kurtenbach et al., 2001; Li et al., 2008b)." (lines 380-383)

"Page 13, Line 370: Since plume G is the only case study from these labels, consider a uniform label for the urban emissions (A) and the remainder of the power plant plumes (B). The increasing lettered format makes it seem that each instance will be discussed."

Response: We have revised Figures 2 and the text, as the referee suggested, and have labeled the plumes according to their sources.

"Page 15, Line 439: The conclusions section of this manuscript is similarly qualitative, as the abstract is, despite the excellent quantitative analysis presented throughout the results and discussion. Suggest revisiting this section with more quantitative information to improve clarity and impact."

Response: We have followed the referee's suggestion, and have revised the conclusions.

"Page 25, Table 2: The +/- SD is in brackets in one part of the table and not the other. Please correct this. The terms PBL and FT are not defined anywhere in the manuscript and should be given at least an operational definition somewhere in the methods section. Lastly, the number of data points being used in each of these calculations should be provided in a column or in the caption."

Response: Revision has made as the referee suggested.

"Page 27, Figure 2: Consider moving this figure to the supporting information or removing it entirely from the manuscript. The only specific features necessary here are the plumes which are presented again in Figure 7. With respect to the urban and power plant plumes, it could be simpler to assign the urban plumes a single letter (such as A), and similarly assign all the power plant plumes with a single letter excepting the one plume discussed in detail, which could be assigned a third letter. With each plume having a different letter, the figure suggests that there is something different between these, when there is nothing in the discussion that suggests this is the case. It would

improve the clarity to simplify this."

Response: The figure has been referred 8 times in the main text. We feel that it is important to keep this figure in the main manuscript so that the reader can get to it quickly. The plumes have been re-labels according to their sources, as the referee suggested.

"Page 28, Figure 3: This figure does not seem necessary for inclusion in the main manuscript and should be considered to be moved to the supporting information. Figure 4 and Table 2 provide redundant and better insight into the measurements."

Response: Again we feel that it is important to keep Figure 3 in the main manuscript; it was referred three times in the discussion. Only a few vertical HONO concentration profiles have been reported so far in literature. They provide important information to understand the budget, the chemistry and the transport of HONO in the troposphere. Figure 3 contains far more data points from 5 research flights over different environments in the southeast U.S., while Figure 4 shows many more parameters from only one race-track over one area, and Table 2 only summarizes the statistics of the measurements. Therefore, they are not really redundant, but rather complementary.

"Page 29, Figure 4: It could be useful to add the typical PBL to FT height as a shaded area (if it has some variability) or horizontal line in each panel to facilitate clarity between the figure data and the discussion."

Response: As stated in the figure caption, Figure 4 shows the vertical distributions of concentrations of HONO, NOx, pNO3, isoprene and potential temperature in the PBL during the first race-track of RF#4. According to the potential temperature and isoprene profiles, the PBL height was around 1200 m.

"Page 30, Figure 5: The two sentences in the paper communicate all the information contained in this figure. Suggest removing this figure altogether or relocating to the supporting information. Further, the correlation analysis undertaken here is unclear

and may be subject to some error if an error-weighted analysis is not being used (Wu and Zhen Yu, 2018). Is the error in both datasets being taken into account when calculating the regression coefficient? Please update the analysis and discussion to reflect the approach and ensure it is robust for the presented data."

Response: The figure has been moved to the supporting information, as the referee suggested. And more robust Deming least-squares regression (Wu and Yu, 2018) has been used in the data analysis, as suggested.

"Page 31, Figure 6: The same regression questions from Figure 5 also apply here. Please clarify the approach utilized and ensure that the appropriate regression analysis and statistics have been used when interpreting the data."

Response: As the reviewer suggested, more robust Deming least-squares regression (Wu and Yu, 2018) has been used in the data analysis.

"Page 32, Figure 7: Panel (a) here can be move to the supporting information or removed altogether."

Response: We have moved the panel (a) to the supporting information as the referee suggested, and have also added $SO_2$ as power plant plume tracer and acetonitrile as a biomass burning tracer to the revised Figure S1, as Andy Neuman suggested in his Short Comments.

"Page 33, Figure 8: This information in this figure is presented concisely in the discussion and the figure does not add anything further. Consider removing this figure from the manuscript."

Response: The figure has been moved to the supporting information as suggested.

"Page 34, Figure 9: The lines are very hard to see on this figure and the green line does not print well. Suggest using two black lines that are thicker than those currently used, with different dashing to distinguish them. The markers are also defined by very thin lines that could be made thicker for clarity."

Response: The figure has been revised as suggested.

References

Gierczak, T., Jimenez, E., Riffault, V., Burkholder, J. B., and Ravishankara, A. R.: Thermal decomposition of HO2NO2 (peroxynitric acid, PNA): Rate coefficient and determination of the enthalpy of formation, J. Phys. Chem. A, 109, 586-596, 2005.

Kurtenbach, R., Becker, K. H., Gomes, J. A. G., Kleffmann, J., Lorzer, J. C., Spittler, M., Wiesen, P., Ackermann, R., Geyer, A., and Platt, U.: Investigations of emissions and heterogeneous formation of HONO in a road traffic tunnel, Atmos. Environ., 35, 3385-3394, Doi 10.1016/S1352-2310(01)00138-8, 2001.

Li, Y. Q., Schwab, J. J., and Demerjian, K. L.: Fast time response measurements of gaseous nitrous acid using a tunable diode laser absorption spectrometer: HONO emission source from vehicle exhausts, Geophys. Res. Lett., 35, 2008.

Neuman, J.A., Trainer, M., Brown, S.S., Min, K.-E., Nowak, J.B., Parrish, D.D., Peischl, J., Pollack, I.B., Roberts, J.M., Ryerson, T.B., and Veres, P.R.: HONO emission and production determined from airborne measurements over the Southeast U.S., J. Geophys. Res.-Atmos., 121, 9237–9250, 2016.

Wu, C., and Yu, J.Z.: Evaluation of linear regression techniques for atmospheric applications: the importance of appropriate weighting, Atmos. Meas. Tech., 11, 1233–1250, 2018 . Ye, C. X., Zhou, X. L., Pu, D., Stutz, J., Festa, J., Spolaor, M., Tsai, C., Cantrell, C., Mauldin, R. L., Campos, T., Weinheimer, A., Hornbrook, R. S., Apel, E. C., Guenther, A., Kaser, L., Yuan, B., Karl, T., Haggerty, J., Hall, S., Ullmann, K., Smith, J. N., Ortega, J., and Knote, C.: Rapid cycling of reactive nitrogen in the marine boundary layer, Nature, 532, 489-491, 2016.

Ye, C., Zhang, N., Gao, H., and Zhou, X.: Photolysis of particulate nitrate as a source of HONO and NOx, Environ Sci Technol, DOI: 10.1021/acs.est.7b00387, 2017.

Please also note the supplement to this comment:
https://www.atmos-chem-phys-discuss.net/acp-2018-105/acp-2018-105-AC4-supplement.pdf

---

## Author Response (AR2)

Dr. Anne Perring
Co-Editor, *Atmospheric Chemistry and Physics*

Dear Dr. Perring,

We are pleased to submit our revised manuscript entitled "Tropospheric HONO Distribution and Chemistry in the Southeast U.S." for publication in *Atmospheric Chemistry and Physics*. We have made the following revisions to the manuscript as you suggested:

(1) Two sentences have been added to address the baseline shifts caused by pressure changes (now lines 168-172): "In addition, large baseline shifts were observed sometimes when the flow state of the scrubbing solution was disturbed by rapid pressure changes during aircraft's rapid ascending to or descending from high altitudes. The data were excluded from analysis if the baseline shifts caused by rapid pressure changes could not be reasonably corrected, regardless of the sign or magnitude of the data."

(2) The two sentences in the original manuscript "The accuracy of HONO measurements was confirmed by comparison with a limb-scanning Differential Optical Absorption Spectroscopy (DOAS) (Platt and Stutz, 2008). The agreement between these two instruments was very good in wide power plant plumes where HONO mixing ratios significantly exceeded the detection limits of both instruments (Ye et al., 2016b)." (lines 147-152) have been revised to "The accuracy of HONO measurements was confirmed by comparison with a limb-scanning differential optical absorption spectroscopy (DOAS) (Platt and Stutz, 2008) during the NOMADSS 2013 summer field study onboard the C-130 aircraft (Ye et al., 2016b). When measuring in wide power plant plumes where HONO mixing ratios exceeded the lower detection limits of both instruments, the agreement between these two instruments was very good, within the assessed uncertainties (Ye et al., 2016b)." (now lines 148-154).

(3) The wording with regard to the Kaser study has been revised, from "the OH radical whose average concentration is estimated at $3 \times 10^6$ mole cm$^{-3}$ in the PBL (Kaser et al., 2015)" (lines263-264) to "the OH radical whose average concentration was found to be ~ $3 \times 10^6$ mole cm$^{-3}$ in the PBL in the Southeast U.S. during the NOMADSS study (Kaser et al., 2015)" (now lines 268-269).

Thank you for your consideration.

Sincerely,

Xianliang Zhou